

# A sixty year ice-core record of regional climate from Adélie Land, coastal Antarctica

S.Goursaud[1,2], V. Masson-Delmotte[1], V. Favier[2,3], S. Preunkert[2,3], M. Fily[3,2], H. Gallée[2,3], B. Jourdain[2,3],
5  M. Legrand[2,3], O. Magand[2,3], B. Minster[1], M. Werner[4]

[1]LSCE (UMR CEA-CNRS-UVSQ 8212-IPSL), Gif-sur-Yvette, France

[2]Univ. Grenoble Alpes, Laboratoire de Glaciologie et Géophysique de l'Environnement (LGGE), 38041 Grenoble, France

[3]CNRS, Laboratoire de Glaciologie et Géophysique de l'Environnement (LGGE), 38041 Grenoble, France

[4]Max-Planck-Institute for Biogeochemistry, Jena, Germany

15  *Correspondence to*: Sentia Goursaud (sentia.goursaud@lsce.ipsl.fr)



**Abstract. .** A 22.4 m-long shallow firn core was extracted during the 2006/2007 field season from coastal Adélie Land. Annual layer counting based on sub-annual analyses of $\delta^{18}O$ and major chemical components was combined with 5 reference years associated with nuclear tests and non-retreat of summer sea ice to build the initial ice core chronology (1947-2007), stressing uncertain counting for 8 years. We focus here on the resulting $\delta^{18}O$ and accumulation records. With an average value of

21.9±6.8 cm w.e. $y^{-1}$, local accumulation shows multi-decadal variations, peaking in the 1980s, but no long-term trend. Similar results are obtained for $\delta^{18}O$, also characterized by a remarkably low (2.6‰) and variable amplitude of the seasonal cycle. The ice core records are compared with regional temperature and stake area accumulation measurements, variations in sea ice extent, and outputs from a high resolution atmospheric general circulation model including stable water isotopes (ECHAM5-wiso) and a regional atmospheric model (MAR), both nudged to ERA atmospheric reanalyses. A significant linear correlation

is identified between $\delta^{18}O$ and regional temperature data, especially in winter. No significant relationship appears neither with regional sea-ice extent nor with Dumont d'Urville wind speed. The model-data comparison highlights the inadequacy of ECHAM5-wiso simulations prior to 1979, possibly due to the lack of data assimilation to constrain atmospheric reanalyses. Systematic biases are identified in the ECHAM5-wiso simulation, such as an over-estimation of the mean accumulation rate and its inter-annual variability, a strong cold bias, and an under-estimation of the mean $\delta^{18}O$ value and its inter-annual

variability. As a result, relationships between simulated $\delta^{18}O$ and temperature are weaker than observed. Such systematic precipitation and temperature biases are not displayed by MAR, suggesting that the model resolution plays a key role along the Antarctic ice sheet coastal topography. Inter-annual variations in ECHAM5-wiso temperature and precipitation accurately capture signals from meteorological data and stake observations, and are used to refine the initial ice core chronology within 1 year. After this adjustment, remarkable positive (negative) $\delta^{18}O$ anomalies are identified in the ice core record and the

ECHAM5-wiso simulation, respectively in 1986 and 2002 (1998-99). Despite uncertainties associated with post-deposition processes and signal-to-noise issues in one single coastal ice core record, we conclude that one single ice core can correctly capture major annual anomalies in $\delta^{18}O$ as well as multi-decadal variations. These findings highlight the importance of improving the network of coastal high resolution ice core records, and stress the skills and limitations of atmospheric models for accumulation and $\delta^{18}O$ in coastal Antarctic areas, particularly important for the overall East Antarctic ice sheet mass

balance.

**Keywords.**

Water isotopes, ice sheet mass balance, decadal climate variability, model evaluation



## 1. Introduction

Deep ice cores from coastal Antarctic areas are crucial to retrieve highly resolved, multi-millennial records, a priority of the International Partnership for Ice Core Science within the Past Global Changes 2k project (Ahmed et al., 2013). Obtaining highly resolved records spanning first the last decades is therefore a pre-requisite to identify the potential of a given site.

Moreover, climate variability of the last decades in Antarctica remain poorly documented and understood (e.g. Jones et al, in press), despite the wealth of information provided by firn and ice cores. This is particularly the case in coastal Antarctic areas, which play a key role for the overall Antarctic mass balance (Agosta et al., 2013; Palerme et al., 2016), with relevance for global sea level (Church et al., 2013). Finally, field data are needed to assess the validity of climate models for multi-decadal variability and change in coastal Antarctic temperature and mass balance (Krinner et al., 2007).

Here, we focus on coastal Adélie Land (Fig. 1), an area where regional climate is marked by the interplay of katabatic winds (Périard and Pettré, 1993), accelerating downslope from Antarctic interior to the coast, a feature known as "the home of the blizzard" (Mawson, 1915), with large seasonal variations in sea-ice extent (König-Langlo et al., 1998; Simmonds and Jacka, 1995) and its proximity to the circumpolar storm track (Jones and Simmonds, 1993). Multi-decadal climate variability in this region is documented since 1958 from Dumont d'Urville (hereafter DDU) meteorological measurements, and since 1979 from

sea ice remote sensing. So far, a few ice cores were drilled near DDU, but none was dated (Jean Jouzel, personal communication), preventing any comparison of the ice core records with meteorological measurements.

In the framework of the International TASTE-IDEA / VANISH programs (Trans-Antarctic Scientific Traverse Expeditions - Ice Divide of East Antarctica), a 22.4 m-long shallow firn core (named S1C1) was extracted in January 2007 from the Adélie Land sector (67.71 °S, 139.83 °E, 279 m a.s.l.) (Fig. 1). The core was collected close to a stake network which provides records

of accumulation spatio-temporal variability from 1971 to 2015 (Favier et al., 2013).

Following the classical approach for other coastal Antarctica areas, the chronology of the S1C1 core is established through the annual layer counting of seasonal cycles in chemical species, water isotopes records and absolute horizons (e.g. Graf et al., 1991; Mulvaney et al., 2002). This task was feasible thanks to a multi-year aerosol monitoring at sub-seasonal scale at DDU providing detailed information on the seasonal cycle of aerosols in relationship with their underlying sources and transport

pathways. However, no such monitoring is yet available for precipitation isotopic composition at this site. Indeed, very few multi-year direct records of snowfall isotopic composition are available from Antarctica (Fujita and Abe, 2006; Landais et al., 2012; Schlosser et al., 2015). And emerging records of water vapour isotopic composition only span one month (Casado et al., 2016; Ritter et al., 2016).We therefore investigated seasonal variations in precipitation isotopic composition from a simulation performed with the isotopically enabled ECHAM5-wiso atmospheric general circulation model (Roeckner et al., 2003; Werner

et al., 2011) nudged to ERA40 (Uppala et al., 2005) and ERA-interim atmospheric reanalyses (Dee et al., 2011). This high-resolution model was chosen due to its good skills with respect to the present-day spatial distribution of water stable isotopes and mass balance in Antarctica (Masson-Delmotte et al., 2008; Werner et al., 2011). The 50 year long simulation is used to identify uncertainties associated with annual layer counting based on the seasonal cycle in precipitation $\delta^{18}O$. Vice-versa, the



ice core record and the stake area network will allow assessing the skills of this simulation in coastal Adélie Land. This model-data is further expanded through investigation of recent mass balance and temperature variations from the regional atmospheric model MAR (Gallée and Schayes, 1994).

Section 2 describes the material and methods underlying this study, including our new measurements, the methodology for the annual layer counting, as well as the regional instrumental (near surface air temperature, wind speed and stake area measurements) and remote sensing (sea-ice extent) datasets, the simulations performed with these two atmospheric models. . Section 3 reports our results. In section 4, we discuss our new ice core records, and compare $\delta^{18}O$ and accumulation records with local to regional climate information as well as atmospheric model outputs. We summarize our main findings and suggest future research directions in our conclusions (Section 5).

## 2.   Material and method

Here, we briefly describe the ice core sampling, the chemical and isotopic analyses, the density measurements and the dating methodology. We then introduce the regional climate datasets used for statistical analyses, as well as the atmospheric simulations used for model-data comparisons.

### 2.1 Field work and ice core sampling

The field campaign was conducted from the 16th to the 26th of January 2007 in Adélie Land by a French joint LGGE (Laboratoire de Glaciologie et Géophysique de l'Environnement) and IPEV (Institut Français Polaire Paul-Emile Victor) expedition. Several shallow firn cores were drilled along a transect between the Italian-French Cap Prud'Homme station, at sea level, and D47 point, located around 1600 m a.s.l. and approximately 100 km from the coast. This round-trip traverse was part of the ANR-VANISH (*Vulnerability of the Antarctic Ice-Sheet*) and IPEV-TASTE-IDEA (*Trans-Antarctic Scientific Traverses Expeditions – Ice Divide of East Antarctica*) scientific programs. The scientific aim of the programs was to obtain new firn core records and ground-penetrating radar measurements to improve the knowledge of coastal East Antarctic accumulation.

The present paper is focused on one ice core drilled at the S1C1 site (66.71 °S, 139.83 °E, 279 m a.s.l.), the closest sampling site to the coast and the French Dumont-D'Urville station (Fig. 1). At this site, the measured temperature of the firn was -13.1±0.5 °C at 10 m-depth. Data obtained during 3 days at 25 cm depth show a mean value of -5.0 °C, assumed to be representative of climatological average summer conditions, based on summer 2007 DDU and Dome C temperatures in a multi-decadal context. Such negative near surface air summer temperature ensures preservation of snow signals (e.g. water stable isotopes, chemical species).

A portable solar powered electromechanical drilling system was used to extract the 22.4 m long and 58 mm diameter firn core (Ginot et al., 2002; Lemeur, submitted; Verfaillie et al., 2012). The recovered core pieces were sealed in polyethylene bags in the field, stored in clean isothermal boxes and transported in frozen state to the cold-room facilities of the Laboratory of



Glaciology (LGGE) in Grenoble, France. Sample preparation for chemical measurements was performed under clean-room laboratory conditions. After stratigraphic observations and measurements of bulk density, the firn core was divided in two half cores. One half was dedicated to radioactivity measurements and the other half was analysed in ionic chromatography and mass spectrometry (Section 2.2). Sub-annual resolution sampling and gross beta radioactivity and gamma spectrometry for

bomb test markers allowed to build the S1C1 firn core chronology (Section 2.3).

## 2.2 Measurements

Discrete density values were calculated based on snow sample volume and mass measurements with a 50 cm resolution, from the surface to 22.4 meters depth. The resulting values range from 400 to 700 kg m$^{-3}$ (Table S2 Supplementary Material). Accounting for measurement difficulties due to weak cohesion of snow in the first few meters, and considering measurement

errors,  the associated uncertainties are ±20-25 kg m$^{-3}$ (± 4%) from the surface to 7 meters depth, and ±15 kg m$^{-3}$ (± 2%) from 7 to 22.4 meters. Hereafter, density values are used to convert snow depths into water equivalent depths.

Firn core sections were processed for artificial radioactivity measurements by low-level (i) beta counting and (ii) gamma spectrometry, with a continuous sampling every 50 cm from the surface to 22.4 m-depth. Using a method developed by Delmas and Pourchet (1977) and improved by Magand (2009), snow samples (100-200 g) were melted, weighed, acidified and filtered

on ion exchange paper, on which radionuclides of interest ($^{90}$Sr, $^{137}$Cs, $^{241}$Am – daughter of $^{241}$Pu) were trapped. All steps were realized in the LGGE clean room laboratory to prevent any contamination. After drying (in an oven at 60 °C), these filters were first analysed by a beta gas-proportional counting system (gross beta measurement) to determine the presence of beta emitters peak(s) corresponding to nuclear weapons global fallout of the 1950s and the 1960s. In a second step, very low level gamma spectrometry (coaxial germanium detector) measurements were performed on the filters to determine activities of $^{137}$Cs

and $^{241}$Am. Both 1955±1 AD and 1965±1 AD peaks were clearly identified in the S1C1 core, respectively at 19.5-20.0 m and 16.5-17.0 m-depth (not shown).

Along the S1C1 core, each 5 cm sample was analysed for ions and the oxygen isotopic ratio. The firn pieces dedicated to the ion analysis were cleaned under a clean air bench located in a cold room (−15 °C) using a pre-cleaned electric plane tool on which the ice is slid (Preunkert and Legrand, 2013). A total of 427 subsamples were obtained along the 22.4 m S1C core, with

a 5 cm sample resolution. Concentrations of cations (Na$^+$ and NH$_4^+$), and anions (Cl$^-$, NO$_3^-$, SO$_4^{2-}$, and methanesulfonate, hereafter MSA), were analysed by ion chromatography (IC) equipped with a CS12 and an AS11 separator column, respectively. Finally, e the oxygen isotopic ratio analysis was processed using the equilibration method on a Finnigan MAT252, using two different internal water standards calibrated to SMOW/SLAP international scales. The accuracy of each measurement is 0.05 ‰ (Fig. 2).



## 2.3 Methodology for layer counting

As shown by Jourdain and Legrand (2002), all ions exhibit a summer maximum at DDU, as also found for different Antarctic coastal stations for nitrate (Wagenbach et al., 1998b), ammonium (Legrand et al., 1998), MSA and sulfate (Minikin et al., 1998). The sodium seasonal pattern at DDU (maximum in summer) is reversed compared to those observed at other coastal

Antarctic sites (e.g. Halley and Neumayer), where the largest concentrations occur in winter, a feature already discussed by Wagenbach et al. (1998a).

The outstanding summer maximum of sea-salt concentrations only detected at DDU is related to its location on a small island, immediately surrounded by open-ocean from December to February. With the aim to use the seasonality of sulfate and MSA produced by dimethyl sulfonic acid emitted in summer by marine phytoplankton, we have calculated the non-sea-salt sulfate

concentrations in the S1C1 core. Wagenbach et al. (1998a) showed that at coastal Antarctic sites in, sea-salt aerosol exhibits a strong depletion of atmospheric sulfate in winter due to precipitation of mirabilite ($Na_2SO_4.10H_2O$) on the sea-ice surface (Wagenbach et al., 1998a). In order to quantify non-sea-salt (nss) $SO_4^{2-}$ contribution, different sulfate to sodium ratio should be defined for summer (0.25, i.e. the reference seawater value) and winter (0.13 derived by Jourdain and Legrand (2002) for DDU):

$[nssSO_4^{2-}]_{summer} = [SO_4^{2-}] - 0.25\ [Na^+]$    (1)

$[nssSO_4^{2-}]_{winter} = [SO_4^{2-}] - 0.13\ [Na^+]$    (2)

Note that the volcanic horizons related to the eruptions of Mt Agung (1963) and Pinatubo (1991) are not detected in the $nssSO_4^{2-}$ profile. Legrand and Wagenbach (1999) explained this feature by the fact that, in coastal Antarctica, the volcanic perturbation remained weak with respect to the year-to-year variability of marine biogenic sulphate. While the relationship

between concentrations in snow and in simultaneously sampled air remains poor for all compounds (Wolff et al., 1998), there is a general seasonal coincidence of the maxima. Annual layer counting is therefore based on our assessment of the concurrence of synchronous maximum values for $Na^+$, MSA, $SO_4^{2-}$, $NH_4^+$, $NO_3^-$ and $\delta^{18}O$ to objectively identify summer horizons. The combination of information on annual layer counting as well as absolute age markers used to build the S1C1 age scale is described in Section 3.1 (Results).

## 2.4 Regional climate records

We extracted DDU monthly near surface air temperature data spanning the period 1958-2007 from the READER data set (https://legacy.bas.ac.uk/met/READER/) (Turner et al., 2004). We also calculated annual mean values of wind speed spanning the period 1950-2007 (with gaps in years 1953-1955) at DDU (P. Pettré, pers. comm.). Finally, we extracted the average sea ice concentration the Nimbus-7 SMMR and DMSP SSM/I-SSMIS Passive Microwave Data (http://nsidc.org/data/nsidc-0051)

and focused on the region situated between 90 °E and 150 °E (Center, 1996). We used these datasets to perform linear regression analysis to explore relationships with the S1C1 $\delta^{18}O$ and accumulation records.



## 2.5 Atmospheric simulations

In this manuscript, we refer to two simulations, one by the global atmospheric model ECHAM5-wiso and one by the regional atmospheric model MAR.

The atmospheric general circulation model (AGCM) ECHAM5 (Roeckner et al., 2003) has been equipped with stable-water isotopes (Werner et al., 2011). This model, named ECHAM5-wiso, leads to realistic simulations when evaluated against observations of isotopic composition in precipitation and water vapour on a global scale. More specifically, the simulated distribution of annual mean precipitation isotopic composition over Antarctica is in good agreement with the spatial isotopic dataset compiled from snow and ice core data (Masson-Delmotte et al., 2008).

In addition to the most common water stable isotope $H_2^{16}O$, $H_2^{18}O$ and HDO were implemented into the hydrological cycle of ECHAM5-wiso, in an analogous manner than in the previous model releases of ECHAM3 (Hoffmann et al., 1998) and ECHAM4 (Werner et al., 2001). Each water phase (vapour, cloud liquid and cloud ice) being transported independently in ECHAM5-wiso contains its isotopic counterpart, implemented in the model code. Equilibrium and kinetic fractionation processes are taken into account during any phase change of a water mass.

The ECHAM5-wiso model simulation analysed in this study covers the period 1960-2007. ECHAM5-wiso was nudged to atmospheric reanalyses from ERA40 (Uppala et al., 2005) and ERA interim (Dee et al., 2011), which was shown to have good skills for Antarctic precipitation (Wang et al., 2016), surface pressure fields as well as vertical profiles of winds and temperature. The ocean surface boundary conditions (sea-ice included) are prescribed based on ERA40 and ERA interim data, too. Isotope values of ocean surface waters are based on a compilation of observational data (Schmidt et al., 2007). The simulation was performed at a T106 resolution (which corresponds to a mean horizontal grid resolution of approx. 1.1 ° x 1.1 °). We used the model results at the grid point closest to our site of interest.

The MAR (Modèle Atmosphérique Régional) model is a limited area coupled atmosphere – blowing snow – snow pack model. Atmospheric dynamics are based on the hydrostatic approximation of the primitive equations (Gallée and Schayes, 1994). The vertical coordinate is the normalized pressure. The parameterisation of turbulence in the surface boundary layer (SBL) takes into account the stabilization effect by the blowing snow flux, as in Gallée et al. (2001). Turbulence above the SBL is parametrized using the local E - ε model of Bintanja (2000). Prognostic equations are used to describe five water species (Gallée, 1995): specific humidity, cloud droplets and ice crystals, rain drops and snow particles. A sixth equation is added describing the number of ice crystals. Cloud microphysical parameterisations are based on the studies of Kessler (1969), Lin et al. (1983), Meyers et al. (1992) and Levkov et al. (1992). The radiative transfer through the atmosphere is parameterised as in Morcrette (2002). Surface processes in MAR are modelled using the Soil – Ice – Snow – Vegetation – Atmosphere – Transfer (SISVAT) scheme (Gallée et al., 2001). In particular, the snow surface albedo depends on the snow properties (dendricity, sphericity and size of the snow grains). Finally the blowing snow model is described in Gallée et al. (2013). MAR is set up over Antarctica with a horizontal resolution of 40 km over 200 x 200 grid points. Lateral forcing and sea surface conditions (SST and sea – ice fraction) are taken from ERA-Interim (Dee et al., 2011). There are 33 levels in the vertical with



a high vertical resolution in the low troposphere. The first level is situated at 3 m a.g.l. The simulation was performed from January 1$^{st}$, 1990, to December 31$^{st}$, 2010.

Hereafter, we use the ECHAM5-wiso simulation to explore the suitability of precipitation $\delta^{18}$O to identify seasonal cycles, to circumvent the lack of regional precipitation monitoring data, as the one-year Dumont d'Urville record is too short to allow

robust analysis. We then compare the ECHAM5-wiso ($\delta^{18}$O, temperature and accumulation) and MAR (temperature and accumulation) outputs with our ice core records.

## 3. Results

### 3.1 Age scale

### 3.1.1 Information from ECHAM5-wiso on annual $\delta^{18}$O cycles

In order to assess whether annual layer counting was possible using $\delta^{18}$O variations, we investigate ECHAM5-wiso outputs to characterize the seasonal cycle of precipitation $\delta^{18}$O in Adélie Land, as well as the variability of the timing of the maximum precipitation $\delta^{18}$O value. The mean simulated precipitation $\delta^{18}$O seasonal cycle over the period 1979-2007 (29 years) shows a maximum in December-January, a minimum in May, and an overall amplitude of 4.45 ‰. The timing of the simulated monthly precipitation $\delta^{18}$O maximum fluctuates from year to year. Out of the 29 years, it was simulated at 80 % during local spring-

summer: 7 times in December (i.e. 24 %), 8 times in November (i.e. 28 %), 8 times in January (i.e. 28 % of the time). However, several occurrences were simulated in autumn: 3 times in March (i.e. 10 %), once in February (i.e. 3 %), and in winter: 2 times in August (i.e. 7 %) (Table S3 Supplementary Material). From this analysis, assuming that the ECHAM5wiso outputs are a perfect representation of reality, peak $\delta^{18}$O has a 80% likelihood to occur in December plus or minus one month. If a chronology was purely based on $\delta^{18}$O , the occurrence of maxima during other seasons may therefore lead to age scale errors of up to 6

months.

### 3.1.2 Absolute age horizons

Following major nuclear tests in the atmosphere, radioactive debris were injected in the stratosphere in 1953-54 and 1963-64 and reported to be deposited in Antarctic in 1955 and 1965 (± 1 year) (Picciotto and Wilgain, 1963), providing absolute dates for two depths in the S1C1 core (red vertical lines, Fig. 2).

Aerosol monitoring data from DDU (1991-2015) show an unequivocal fingerprint of rare summers during which the sea-ice retreat was not complete offshore the site, through the absence of sodium maximum observed in January 1995 (Jourdain and Legrand, 2002), and more recently in January 2012 and 2013 (Legrand et al., 2016). Prior to the start of DDU atmospheric monitoring in 1991 and to the availability of satellite sea-ice data in January 1979, two other years were characterized by a non-complete sea-ice retreat in summer offshore DDU, namely January 1969 and January 1979 (Patrice Godon, personal

communication). These years were used as absolute horizons in the ice core dating (purple vertical lines, Fig. 2).



### 3.1.3 Combining layer counting with absolute age horizons

We assessed the concurrence of synchronous maximum values for $Na^+$, MSA, $SO_4^{2-}$, $NH_4^+$, $NO_3^-$ and $\delta^{18}O$ to objectively identify summer horizons, indicated by the vertical continuous black lines in Fig. 2. Only 4 layers showed simultaneous peaks in all species (in 1959, 1966, 1984 and 2001). In the other cases, the determination was equivocal: either data were missing for the depth interval corresponding potentially to summer (4 times for MSA and 2 times for $\delta^{18}O$), or peaks were not identified in some of the records (4 times for $Na^+$, 10 times for MSA, 4 times for the $nssSO_4^{2-}{}_{,summer}$, once for $NO_3^-$, 7 times for $NH_4^+$ and 25 times for $\delta^{18}O$), or, conversely 2 or 3 peaks appeared (9 times for $Na^+$, 2 times for MSA, 5 times for the $nssSO_4^{2-}{}_{,winter}$, 6 times for the $nssSO_4^{2-}{}_{,summer}$, 7 times for $NO_3^-$, and 5 times for $NH_4^+$). We note that maxima in $NO_3^-$ and $\delta^{18}O$ were often delayed with respect to maxima in other species. These two parameters can potentially be affected by diffusion in the firn, due to snow-vapor interactions (Johnsen, 1977; Mulvaney et al., 1998). Mulvaney et al. (1998) also reported that MSA could migrate in the firn. Potential seasonal signals were generally associated with a very low amplitude, especially from 1969 to 1975, or no signal could be discerned. Altogether, $nssSO_4^{2-}{}_{,winter}$ seems to be the most reliable species to identify seasonal cycles, as a seasonal pattern appears for every layer.

The remaining difficulty is associated with multiple peaks within one year. This appears for instance in year 1993 (represented in dashed vertical line in Fig. 2), for which double peaks may be associated with year 1993 only, or with two successive years. Such situations were encountered 8 times (dashed vertical lines in Fig. 2). As a result, we made subjective choices for the identification of annual or sub-annual peaks in order to optimize the coherency with absolute age markers. Indeed, 68 potential annual layers were initially identified, and this result is not compatible with the horizons based on these two sources of information (radionuclide peaks and years without summer sea-ice retreat). We then repeated the annual layer counting, excluded layers for which we had the smallest number of maxima among the species.

We found 60 summer peaks represented in Fig. 2 with solid lines, whereas the initial layers excluded from the final chronology are represented with dashed lines. As a result, the S1C1 ice core spans the period 1947-2007, as displayed in a depth (expressed in meter water equivalent)-age scale (Fig. 3).

### 3.2 Record of accumulation variability and comparison with stake measurements

Annual accumulation variations were extracted from the S1C1 core by combining density measurements with annual mean layer counting (Fig. 4). The resulting annual accumulation varies between 9.7 and 38.0 cm w.e. $y^{-1}$, with a mean of 21.9 cm w.e. $y^{-1}$, and a standard deviation of 6.8 cm w.e. $y^{-1}$. This value is 14% lower than the annual accumulation of 25.6 cm w.e. $y^{-1}$ (standard deviation of 15.1 cm w.e. $y^{-1}$) measured between 2004 and 2015 on one stake located closest to the drilling site (Agosta et al., 2012) (Fig. 1b), or 27% lower than the mean value of 29.9 cm w.e. $y^{-1}$ (standard deviation of 26.6 cm w.e. $y^{-1}$) measured nearly at the same location between 1971 and 1991. We identified remarkable years as those for which accumulation deviates from the mean value by more than two standard deviations. Remarkable high accumulation rates were thus encountered in 1981, 1983 and in 2001 (at respectively 35.9, 37.3 and 38.0 cm w.e. y-1). No remarkable low accumulation





rates were found. For stake accumulation measurements performed at the stake closest to S1C1, the highest values were measured in 1979 and 1981 (75.8 and 73.8 cm w.e. $y^{-1}$). None of these remarkable years can be found neither in the stake average record nor in the stake data the closest to the S1C1 site. Moreover, small and non-significant linear correlation coefficients are found with the stake average record (r=0.15 and p=0.6), and with the stake data the closest to the S1C1 site (r=0.38 and p=0.2). There is no significant long term. Spectral analyses performed using the Multi-Taper method (Paillard et al., 1996) reveal significant periodicities of 5 and 8 years (harmonic F-test ≥ 0.94). We also note a periodicity of 12 years, but which is not detected as significant (harmonic F-test=0.8). Decadal variability is characteristic of the high latitudes of the Southern Hemisphere (Yuan and Yonekura, 2011). This motivates filtering using a 13-point binomial filter with bounce end-effects (Fig. 5) , i.e. repeating the first (last) value six times in order to calculate the six first (last) running means (Fig. 5).From this analysis, we observe a slight accumulation increase from the 1950s until the 1980s, with secondary maxima in 1970s and 1990s (Fig. 5). A decrease appears from the 1980s to the 2000s, back to the initial mean level. Finally, the S1C1 core displays an unprecedented accumulation increase from the 1990s to the 2000s.The latter feature is surprising, as recent stake measurements from this area show no increase in mean values observed in 2004-2015 compared with earlier surveys (1971-1991). We therefore remain cautious about the recent accumulation increase inferred from one single ice core record, unless this finding can be supported by other lines of evidence. Finally, note that only a 5 year periodicity emerges from spectral analysis of DDU temperature (harmonic F-test=0.99).

### 3.3 Record of $\delta^{18}O$

With our age scale, each year is documented with 2 to 14 $\delta^{18}O$ 5 cm measurements, with a mean number of 7 values per year. Hereafter, the $\delta^{18}O$ record is resampled at this average temporal resolution using the Fourier method (Fig. 4). Annual mean $\delta^{18}O$ values are then calculatedfrom January to December, for comparison with accumulation and other observational or simulation datasets (Fig. 5).

### 3.3.1 Other sources of information for $\delta^{18}O$ regional variability

Very few multi-year direct records of snowfall isotopic composition are available from Antarctica (Fujita and Abe, 2006; Landais et al., 2012; Schlosser et al., 2015), and none in our region of interest. Near S1C1, the single direct observation arises from a collection of 19 samples of precipitation at DDU, spanning the period from February to November of 1977 (*J. Jouzel, pers. comm.*). These data depict a mean $\delta^{18}O$ value of -18.0 ‰, a standard deviation of 3.9 ‰, with the lowest value in August (~ -27.7 ‰) and the highest values in March (-10.0 ‰) and November (-11.5 ‰). Despite its brevity, this dataset shows less depleted values, a different seasonal pattern, and lower variability than at Dome C, where recent measurements are available for the period 2008-2010 (Stenni et al, The Cryosphere, submitted) ranging from -80.6 to -35.5 ‰, with a ~37 ‰ seasonal cycle in phase with that of near surface air temperature. The distribution of $\delta^{18}O$ data at DDU appears consistent with the range of values observed in a multi-decadal (1982-1996) record of surface snow isotopic composition obtained from the coastal station of Neumayer, with a mean $\delta^{18}O$ value of -20.6 ‰, and a seasonal amplitude varying from 14.6 to 29.0 ‰ (Schlosser



et al., 2008). Finally, the Dumont d'Urville precipitation mean $\delta^{18}O$ value is also quite close to that of shallow ice cores from Law Dome (66.73 °S, 112.83 °E, 1395 a.s.l.) (Delmotte et al, 2000; Masson-Delmotte et al, 2003), which display a mean value of ~ -21 ‰, and an average seasonal $\delta^{18}O$ amplitude of ~ 6 ‰, together with a larger inter-annual variability rather in local winter than in local summer.

### 3.3.2 Record of $\delta^{18}O$ variability and link with accumulation and regional climate records

The sub-annual record from raw measurements and resampling at an average of 7 points per year displays large year-to-year variations in the amplitude of the seasonal cycle, varying from 0.2 ‰ (in 1980) to 12.5 ‰ (in 2005), with a mean value of 2.5 ‰. The S1C1 record clearly displays a smaller intra-annual range than observed in DDU precipitation (17.7 ‰ for one single year), and an average intra-annual range about twice smaller than shallow ice core records from Law Dome (typically 5 ‰). Note that however, confidence in this finding is limited by the resolution of our record as well as by potential post-deposition processes affecting the preservation of the precipitation isotopic composition signal (Touzeau et al., 2016).

We again identified remarkable years as those deviating from the mean value by more than two standard deviations. Remarkable low $\delta^{18}O$ values were encountered in 1967, 1997 and 2005, while remarkable high $\delta^{18}O$ values were encountered in 1971, 1985 (highest value), 2001, and 2006. The year 1997 is marked by the most depleted annual mean $\delta^{18}O$ values of the whole series. No long-term trend is observed. Spectral analyses performed using the Multi-Taper method evidence highly significant periodicities of 6 and 11 years (harmonic F-test > 0.99), close to those identified for accumulation. . This former result lead us to use a 13-point binomial filter with bounce end-effects to explore the S1C1 $\delta^{18}O$ record at the decadal scale and to compare it with accumulation at the same scale. The resulting calculations depict a sharp decrease in the 1950s, with the strongest decadal minimum peaking in 1955. It is followed by a gradual increase, reaching the highest $\delta^{18}O$ decadal-mean values in ~1982 and ~1992, with secondary maxima in ~1962 and ~1972. A sharp decrease occurs in the 1990s, when mean levels from the 1950s are reached again. There is an anti-phase of the $\delta^{18}O$ and accumulation smoothed signals in the 1950s and the 2000s, while they appear in phase from the 1970s to 1990s, with multi-decadal variations.

We now explore linear relationships between $\delta^{18}O$ accumulation records and regional climate records (Table 2). No significant linear correlation is identified between annual mean S1C1 $\delta^{18}O$ and accumulation records from 1947 to 2006, but a significant positive linear relationship appears from 1997 to 2006. We also note that remarkable years for $\delta^{18}O$ and accumulation do not coincide.

There is no linear relationship between annual mean S1C1 $\delta^{18}O$ and regional records of near surface air temperature over the whole available period. However, a linear correlation appears since 1979, displaying a slope of 1.14 ‰ per °C with the strongest and more significant correlation coefficient during local winter, whereas it is the weakest (and not significant) relationship during local summer. At the decadal scale, the correlation between $\delta^{18}O$ and the near surface air temperature is much stronger, with a linear slope of 1.49 ‰ per °C. At Law Dome, the DE08 ice Australian $\delta^{18}O$ record displays a weak correlation with Casey near surface air temperature with a slope of -0.31 ‰ per °C (r=-0.45 and p=9.0E-3 for theperiod 1959-1991) (Masson-Delmotte et al., 2003). At Dome C, over 3 years only, annual mean precipitation $\delta^{18}O$ displays a significant





correlation with AWS near surface air temperature with a slope of 0.49 ‰ per °C (r=0.79 and p=2.5.E-110. By comparison, the $\delta^{18}$O-temperature slopes found for S1C1 (Table 3) are particularly high compared with previous results from Law Dome ice cores, and from Dome C precipitation data.

No significant relations are detected between annual mean S1C1 $\delta^{18}$O and annual mean sea ice concentration in the Adélie land sector, and between annual mean S1C1 $\delta^{18}$O and annual mean wind speed over the whole respective available periods. Again, higher correlation coefficients have emerged since 1979 and 2000 for the sea ice concentration in the Adélie Land sector and the wind speed, respectively.

### 3.3 Model-data comparison

### 3.4.1 Comparison between the ECHAM5-wiso and the S1C1 data

This comparison is limited by the fact that we only have one firn core record, and no estimation of the deposition noise, which can only be estimated from signal-to-noise analyses of multiple ice core records from the same area (Masson-Delmotte et al., 2015). Moreover, recent monitoring of surface snow and surface vapour isotopic composition in polar regions have suggested isotopic exchanges in-between snowfall events (Ritter et al., 2016; Steen-Larsen et al., 2014), in relationship with surface snow metamorphism. Such processes are not accounted for in the atmospheric models, and we can therefore only compare simulated precipitation-weighted $\delta^{18}$O with our firn measurements. Similarly, we compare the ECHAM5-wiso model simulated precipitation (or precipitation minus evaporation) with the ice core accumulation data, which may also reflect the impact of wind erosion.

The ECHAM5-wiso simulation produces a large increase in $\delta^{18}$O and precipitation after 1979 (not shown). As such increase is not observed in the S1C1 record, we suggest that it may arise from a discontinuity in the ERA reanalyses data used as boundary conditions. Indeed, the lack of sea ice observations prior to the satellite era in the ERA-40 data set leads to the unrealistic set of climatological sea ice coverage values around Antarctica prior to 1979. This deficit does not exist in the ERA interim data set, starting from year 1979. Therefore, in the following, the model-data comparison is restricted to the period 1979-2007 (Fig. 6).

The ECHAM5-wiso simulation underestimates the average seasonal amplitude of $\delta^{18}$O at the S1C1 grid point by a factor of 2.0, and its inter-annual standard deviation by a factor of 1.4. No strong similarity appears for year-to-year values. Neither the S1C1 record nor the model outputs present any long-term trend (r=-0.48 and p=8.0E-3 and r=-0.38 and p=4.0E-2 respectively). By contrast, the model precipitation amount (68.4 ±11.9 cm w.e. $y^{-1}$) is much higher than observed (23.1 ±7.3 cm w.e. $y^{-1}$) at S1C1 site on the same period. This difference cannot be explained by the impact of sublimation (in average, -4.8 cm w.e. $y^{-1}$ in the simulation). Note that the model may not fully account for the impact of wind erosion, due to missing parameterizations for wind scouring and misrepresentation of small-scale katabatic winds. However, snow accumulation is highly variable in this region, due to the orographic effect on precipitation and to wind scouring. Along the first 50 km of the stake network (Agosta et al., 2012), the mean accumulation rate is estimated at 34.6 cm w.e. $y^{-1}$ between 2004 and 2014. This result confirms



that ECHAM5-wiso overestimates the regional accumulation rate. When considering deviations from average values (Fig. 6), no similarity emerges between inter-annual variations in the S1C1 record and in the simulation. The simulation captures the observed positive anomalies in 1993. No significant long-term trend is identified from 1979 to 2007 (r=0.12 and p=0.56 for S1C1 and r=-0.31 and p=0.1 for ECHAM5-wiso).

The model-data mismatch may arise from the topographic effect of the Antarctic ice sheet margin on the distribution of precipitation, and the associated isotopic distillation. The model grid size used here (approx. 1.1 ° x 1.1 °, i.e. 110 x 110 km$^2$) might also be too coarse for a direct comparison of the simulation results to the S1C1 drill site. Further investigations are needed, and this requires also a regional network of shallow firn core records on the Antarctic ice sheet margin, in order to characterize spatio-temporal variations and signal-to-noise ratios.

In the simulation, we have also explored the $\delta^{18}$O-precipitation and the $\delta^{18}$O-temperature relationship. A significant linear relationship appears for temperature, with a slope of 0.32 ‰ °C$^{-1}$ (r=0.42, p= 2.0E-2), therefore much smaller than calculated from the S1C1 core and the DDU temperature data. By contrast, no significant linear relationship is simulated for precipitation (r=0.22, p=0.2).

   In summary, the ECHAM5-wiso model outputs appear to have a wet and $\delta^{18}$O-depleted bias at the S1C1 drill site, overestimate

the inter-annual accumulation variability, underestimate the $\delta^{18}$O variability and its relationship with surface air temperature, and fail to capture the observed $\delta^{18}$O-accumulation relationship.

   Our model-data comparison is of course limited by comparing two different data sets. For the modelling results, we have analysed a pure atmospheric signal (precipitation). In the ice core datasets, we have looked at a signal which is produced by precipitation, but also potentially altered by post-deposition processes such as sublimation, wind scouring and snow

metamorphism (which may affect the isotopic signal). Also, the fact of having just one ice core record (also associated with age scale uncertainties) precludes an assessment of the signal-to-noise ratio. Indeed, with similar mean accumulation values, it was shown that 4-5 shallow ice cores may be needed to extract the common climatic signal in North-West Greenland (Masson-Delmotte et al, 2015).

### 3.4.2 Comparison between the ECHAM5-wiso model and the MAR model

We now investigate the results of the regional model MAR for accumulation. MAR produces a more realistic range of accumulation, with simulated values varying from 12.3 cm w.e. y$^{-1}$ (in 2006) to 32.4 cm w.e. y$^{-1}$ (in 1999). At this 40 km horizontal (against 110 km in the ECHAM5-wiso model), the slope of the East Antarctic ice sheet is better represented, a critical aspect for orographic precipitation. However, the inter-annual variability simulated by MAR is very different from that inferred from the S1C1 core compared to ECHAM5-wiso. This result may arise from the nudging technique: while the

ECHAM5-wiso model is strongly nudged to reanalyses, the MAR model is only forced by large-scale circulation at the lateral boundaries and layers above 20 km, which allows the model to build its own atmospheric circulation. We conclude that systematic caveats of the ECHAM5-wiso model for coastal Antarctic precipitation amounts may arise from its resolution.





## 4.   Discussion

Here, we propose an alternative ice-core chronology, given the annual layer counting uncertainties. For that purpose, we have screened peak-to-peak relationships between S1C1 record, stake accumulation measurements, and ECHAM outputs. We first tried to optimize the match between S1C1 accumulation and stake data. However, the inter-annual standard deviation of S1C1
accumulation is approximatively three times lower than calculated from either individual stake records, or from the average stake record. As a result, we could not identify unequivocally remarkable low or high accumulation years as a robust signal. For instance, all stake area data show very low (close to zero) accumulation in years 1972 and 1988, but no such signal can be identified in the S1C1 record. We therefore used the ECHAM5-wiso $\delta^{18}$O to test an alternative chronology (Section 4.1). We identified a high similarity between ECHAM5-wiso inter-annual precipitation variability and average stake data (Section 4.2).
We then discussed processes that can produce non climatic noise (Section 4.3). After assessing $\delta^{18}$O-temperature relationships (Section 4.4), we finally explored the spatial relevance of climatic records from the S1C1 location (Section 4.5).

### 4.1 An alternative age scale can improve the model-data comparison for $\delta^{18}$O

The comparison of the most remarkable annual anomalies in the S1C1 and simulated $\delta^{18}$O series from ECHAM5-wiso reveals a one year shift: the minimum value is identified in 1997 for S1C1 but simulated in 1998, while the 1985 maximum value in
the S1C1 core may coincide with the simulated maximum value in 1985. Given the uncertainties of age chronology of at least one year and potentially several years (Section 3.1), we reconsidered the annual layer counting to assess whether the S1C1 chronology can be reconciled with the ECHAM-wiso $\delta^{18}$O outputs.

Indeed, the identification of the summer 2001-2002 in our initial chronology (year 2002 in Fig.2) is based on equivocal evidence: no peak of MSA, small peaks of both $nssSO_4^{2-}{}_{winter}$ and $nssSO_4^{2-}{}_{summer}$ (at a magnitude not systematically counted as
a summer). Vice-versa, we could have missed one year between the initial layers associated with the summers 1981-1982 and 1982-1983, where peaks are identified in $nssSO_4^{2-}{}_{winter}$, $nssSO_4^{2-}{}_{summer}$, $NO_3^-$ and $NH4^+$. We therefore built an alternative depth-age scale (Table S4 Supplementary Material) following these two findings (Fig. 7 and Fig. 8), and discussed the implications for the S1C1 records.  Consequently, we have produced new annual mean accumulation and $\delta^{18}$O time series, from 1979 to 2007, following the same approach (Section 2.3.3).

With this alternative chronology, the resulting mean accumulation does not change, but the inter-annual standard deviation is enhanced by 44% ($23.1 \pm 9.8$ cm w.e. $y^{-1}$).  It has no impact on $\delta^{18}$O or accumulation multi-decadal variations or trends. The resulting accumulation dataset remains non-significantly correlated with the stake average record and with the closest stake data. The alternative chronology does not improve neither the strength of the $\delta^{18}$O-temperature relationship between the S1C1 record and the DDU near surface air temperature time series (r=0.49 and p=1.0E-2) (but slightly increases the slope of this
relationship with a new value of 1.24 ‰ °$C^{-1}$), nor the relationship between S1C1 accumulation dataset and the ECHAM5-wiso precipitation output (r=0.19 with p=0.7 for the initial chronology, and r=-0.08 with p=0.3 for the alternative chronology).





However, it improves (by construction!) the strength of the correlation between the S1C1 $\delta^{18}$O and the ECHAM5-wiso $\delta^{18}$O, from r=0.44 with p=2.0E-4 for the initial chronology to r=0.64 with p=2.0E-2 for the alternative chronology

### 4.2 Comparison between stake data and ECHAM5-wiso simulation

No significant linear correlation is found neither between the S1C1 accumulation record (whatever chronology is used) and the stake accumulation measurements, nor between the S1C1 accumulation record and the ECHAM5-wiso simulated precipitation. We therefore compared the stake measurements and the ECHAM5-wiso output, with the goal to investigate if there is a common climatic signal (for theperiod 1979-1991).

We observe a high and significant correlation between the ECHAM5-wiso simulated precipitation and the average stake record at the inter-annual scale (r=0.62 and p=3.0E-2). This finding shows that the nudged ECHAM5-wiso simulation is able to capture almost 40% of the inter-annual average accumulation variance, and that there is a clear climatic deposition signal in the stake accumulation record. We also note that there is a high dispersion in the inter-annual variability of accumulation across the individual stake records, as evidenced by the quartile distribution (Fig. 7). Each stake record is significantly correlated with the average stake record (r varies from 0.57 to 0.90, reaching 0.78 for the stake closest to S1C1, p=3.2E-5). However, significant differences are encountered. For instance, the sharp peak in accumulation displayed by the stake closest to S1C1 in 1979 is an outlier and is not recorded in the average stake signal. Similarly, the correlation coefficient between the ECHAM5-wiso simulated precipitation and individual stake records varies from 0.3 to 0.7, stressing potential non-climatic noise in local accumulation data.

We conclude that the common climatic signal in average stake accumulation time series is related to regional-scale circulation, and is captured by the ECHAM5-wiso simulation. The lack of correlation between the S1C1 accumulation record and both stake records and ECHAM5-wiso precipitation outputs may therefore reflect local non-climatic noise.

### 4.3 Processes causing non-climatic noise

The significant correlation between the ECHAM5-wiso precipitation and the stake average accumulation record stresses the fact that the inter-annual variability of the average accumulation is dominated by large-scale atmospheric circulation controls on precipitation. Given the spatio-temporal variability of accumulation reported previously, we could not use accumulation information to refine the initial S1C1 chronology, and conclude that the S1C1 accumulation record is probably heavily affected by non-climatic noise. The significant correlation between the ECHAM5-wiso precipitation and the stake average accumulation record stresses the fact that the inter-annual variability of the average accumulation is dominated by large-scale atmospheric circulation controls on precipitation. However, snow accumulation is spatially variable at the study site (Agosta et al., 2012), associated to wind erosion, wind redistribution, sublimation, occurrence of gravity waves in the katabatic flow, and other processes during or after the precipitation event (Eisen et al., 2008). This leads to very high variability even at very small spatial scales, down to deca-meter scale (Libois et al., 2014), as reflected by the presence of sastrugi (Amory et al., 2015). Local features associated either with the distribution of precipitation or with wind-driven post-deposition erosion or



deposition appear dominant for a given stake record, and for the S1C1 record as well. Differences in accumulation estimates from the stake data and from S1C1 may also arise from variations in surface snow density. Indeed, stake height measurements were converted into water equivalent using a spatial estimate of mean density distribution (Agosta et al., 2012). Lastly, his analysis also highlights the limited information which can be extracted from one single ice core. In the future, obtaining several

ice cores from this sector will help to quantify the common climatic signal and better characterize the non-climatic noise, and to confirm our hypothesis that there is a significant climatic signal in $\delta^{18}$O from this single core, as suggested by the comparison with the ECHAM5-wiso output.

## 4.4   $\delta^{18}$O-temperature relationship

We now report the various contradictory estimates of the $\delta^{18}$O-temperature relationship based on our S1C1 record, instrumental

temperature data, and ECHAM5-wiso outputs (Table 3).

Correlations and slopes in the $\delta^{18}$O-temperature relationship from the data are higher at the decadal scale compared to the annual scale. It suggests that the $\delta^{18}$O-temperature is not stationary through time, and that the calibration established at the inter-annual scale under-estimates the strength of the relationships at the decadal scale.

This may arise from non-climatic noise at the inter-annual scale. Moreover, the decadal slopes are stronger than expected

from a Rayleigh distillation, possibly resulting from either a decoupling between surface and condensation temperature, or from changes in moisture sources through time.

The ECHAM5-wiso simulation (Section 2.3.1) shows that the seasonal cycle of $\delta^{18}$O is not in phase with the seasonal cycle of temperature. Moreover, the inter-annual variability of $\delta^{18}$O simulated in the S1C1 area is maximum in local spring (November), in contrast with the maximum inter-annual variability of DDU temperature, occurring in winter. This finding are also different

from Dome C precipitation data (Stenni et al, CPD, submitted), showing a larger inter-annual variability of both temperature and $\delta^{18}$O during local winter (possibly due to variability in frequency of maritime air mass intrusions). These model results suggest that processes other than local temperature play a key role in local $\delta^{18}$O seasonal cycle and inter-annual variability.

The $\delta^{18}$O – temperature relationship within the ECHAM5-wiso model is lower by a factor of 2 to 3 than inferred from the S1C1 records. To understand this mismatch between data and simulations, we investigated alternative approaches for the

$\delta^{18}$O-temperature relationship obtained using annual data through the combination of temperature extracted from the READER database and simulated $\delta^{18}$O, and the temperature extracted from the ECHAM5-wiso model and S1C1 $\delta^{18}$O record (Table 3). We found a high and significant correlation between the temperature from the READER database and ECHAM5-wiso simulated near surface air temperature but a significant and weaker correlation between $\delta^{18}$O from the S1C1 core and simulated by ECHAM5-wiso. The variance of the ECHAM5-wiso model result is about twice smaller than in the S1C1 core (inter-annual

standard deviation of 0.6 ‰ and 1.8 ‰ for the ECHAM5-wiso model and in the S1C1 core respectively (Table 1). We conclude that the low simulated $\delta^{18}$O-temperature relationship arises from a much smaller variability in the simulated $\delta^{18}$O than in the S1C1 record.



So far, we cannot conclude on the cause for such a difference in variance: the ice core record may be affected by post-deposition features that generate noise, or the atmospheric model may not adequately resolve moisture transport pathways driving variability in the precipitation signal. Systematic comparisons between ECHAM5-wiso and results from direct precipitation sampling or from surface water vapour isotopic composition monitoring available will be needed to understand whether this mismatch arises from the atmospheric model or from signal-to-noise issues in one individual ice core record. So far, no such measurements are available in Adélie Land (apart from precipitation data from 1977 reported in our introduction, but in a time period when no sea ice information is available and reanalyses are less reliable). In any case, this finding calls for a cautious use of simulated $\delta^{18}$O-temperature relationships in coastal areas, before a full assessment of the model skills can be achieved.

### 4.5 Spatial extent of the representativeness of S1C1 site

We finally take advantage of the coherent framework provided by the ECHAM5-wiso simulation in order to investigate the spatial relevance of the S1C1 site. For that purpose, we have calculated the correlation coefficients of the annual surface air temperature, precipitation amount and $\delta^{18}$O between the model output at the S1C1 site and the model outputs for each other Antarctic grid point (cf. Fig. 9). Regarding temperature, S1C1 temperature appears closely related ($r \geq 0.8$) with regional temperature in an approximative ~1000 km from ~120 °E to 150 °E. S1C1 is strongly correlated with regional precipitation amount, although for a much reduced area close to S1C1, within ~500 km. In the model world, S1C1 precipitation-weighted $\delta^{18}$O appears only related with regional signals, with the strongest correlation coefficients for a very local area (approximatively 100 km width).

### 5. Conclusions and perspectives

Documenting inter-annual climatic variations in coastal Antarctica is important to characterize natural climate variability, and provide a long-term context for recent changes, beyond the instrumental period. The study of the S1C1 core adresses this challenge.

The initial chronology of the S1C1 ice core was established combining multi-parameter annual layer counting using major ions and $\delta^{18}$O, as well as references horizons, including nuclear test horizons and fingerprints of summers with no sea-ice retreat. The latter approach would not have been feasible without the knowledge provided by long-term aerosol monitoring at DDU. The S1C1 records encompass the period 1947-2007, and we have provided a clear description of sources of age scale uncertainty.

The mean accumulation rate estimated from the S1C1 record is coherent with information from stake area measurements in Adélie Land. No long-term trend is detected in the stake data (Agosta et al., 2012) and in the S1C1 accumulation record. This robust finding challenges a recent regional modeling study showing a decrease of accumulation in coastal Adélie Land from 1979 to 2010 (Lenaerts et al., 2012). The record is marked by periodicities of 5, 8 and 12 years, and multi-decadal variations, peaking in the 1980s. We are not confident in the signal of higher accumulation rates in the 2000s, because no such signal is



detected in the comparison of recent and historical stake measurements. The average stake area record is significantly correlated with results from the ECHAM5-wiso model, but not with the S1C1 record. This finding may arise from post-deposition noise associated with e.g. wind scouring in an area marked by remarkable katabatic winds. The lack of similarity between the S1C1 ice core records and the sake measurements may also arise from the fact that the location of stake measurements may have moved through time (Agosta et al., 2012), and from the correction of height measurements using an average density value along the profile rather than simultaneous measurements of density which were not available.

The S1C1 $\delta^{18}O$ record shows a remarkable small amplitude of the seasonal cycle, when compared to other isotopic datasets from coastal East Antarctica. We have evidenced a relationship between inter-annual variations of $\delta^{18}O$ and accumulation, but stressed that remarkable years are not coincident. Both records display a multi-decadal variability. The S1C1 $\delta^{18}O$ record is significantly correlated with DDU near surface air temperature, with a particularly large slope. We stress that the ECHAM5-wiso model strongly underestimates the inter-annual variability in $\delta^{18}O$ and therefore the strength of the slope of the $\delta^{18}O$-temperature relationship. The under-estimation of the inter-annual variance of precipitation $\delta^{18}O$ by the ECHAM5-wiso model calls for a systematic assessment of the skills of the isotopically enabled atmospheric general circulation models against the whole variety of data regarding the temporal variability (vapour, precipitation and firn isotopic composition using shallow ice core data as well as the few precipitation and vapour measurements). In order to better understand the drivers of isotopic variability in coastal Adélie Land, obtaining continuous vapour isotopic composition measurements at DDU will offer the possibility of a direct comparison with atmospheric transport pathways, and with aerosol chemical data.

Implementing water stable isotopes in the regional MAR model is underway. Resolving regional aspects of the atmospheric circulation, especially related to katabatic winds and boundary layer dynamics, may indeed be important to resolve changes in moisture origin at the inter-annual scale. Contrary to the regional MAR model, the ECHAM5-wiso model captures inter-annual variations in Adélie Land accumulation rate (as inferred from the average stake measurements) and remarkable $\delta^{18}O$ anomalies (as identified in the S1C1 record). Therefore, these variations are driven by large-scale atmospheric circulation. We have highlighted the fact that model outputs cannot be used in a climate sense prior to 1979, probably due to the lack of sea ice data used in the assimilation systems.

In this manuscript, we have also used the ECHAM5-wiso model outputs to challenge the interpretation of ice core $\delta^{18}O$ records. First, we have quantified the source of errors associated with the use of seasonal cycle in precipitation $\delta^{18}O$ to detect summer layers. Second, we have identified the two strongest anomalies in the simulated and S1C1 $\delta^{18}O$ records, and challenged our initial ice core chronology. None of our key conclusions is affected by the alternative chronology implying a one-year shift, and compatible with the multi-parameter information used for layer counting. Third, we have explored the spatial relevance of precipitation and precipitation $\delta^{18}O$ outputs at the S1C1 site, and stressed the differences between a large-scale representativeness of temperature (typically, 1000 km), and a smaller-scale representativeness of $\delta^{18}O$ (typically, 100 km around the S1C1 site). In the future, obtaining a network of shallow ice core records in coastal East Antarctica will be critical to assess signal-to-noise aspects and test the validity of these findings. Obtaining longer records will also allow to better characterize the multi-decadal variability and assess past changes in Antarctic temperature (Ahmed et al., 2013; Jones and al.,



in press). Finally, characterizing spatio-temporal variations in deuterium excess will provide a complementary line of information, for the identification of seasonal variations, and for assessing past changes in moisture origin.

Further investigations are finally needed to explore the relationships between large-scale modes of variability such as the Antarctic annular mode, the Pacific – South American patterns, ENSO and climate variability in the Adélie Land sector.

5 Indeed, composite analyses performed from 1979 to 2013 show significant imprints of SAM and PSA2 on DDU temperature (Marshall and Thompson, 2016), and of La Niña on coastal East-Antarctic climate (Welhouse et al., 2016). During the same time period, S1C1 $\delta^{18}O$ (using the second dating) is significantly anti-correlated with the SAM index (r=-0.43 and p=2E-2), and significantly correlated with the Niño3.4 index (r=0.50 and p= 5.5E-3). Longer records with multiple ice-cores will allow to test the stability of these teleconnections through time (Yeo and Kim, 2015).

10 We have evidenced a multi-decadal variations in the S1C1 ice core records, DDU near surface air temperature and sea-ice extent. Our record, available back to 1947 (therefore 10 years earlier than the instrumental temperature series from DDU) depicts a decadal $\delta^{18}O$ minimum in the mid-1950s, within the range of subsequent variability. Altogether, these results imply no major climatic reorganization in this sector of East Antarctica prior to the 1970s. It contradicts suggestions for dramatic sea ice contraction in the mid-1950s, based on whaling ship records (Cotte and Guinet, 2007; William, 1997). Given discussions 15 with respect to drivers of penguin demography (Raymond et al., 2015; Southwell et al., 2015), reconciling all sources of information for Adélie Land climate variability prior to the 1970s remains to be achieved.

**Author contribution.** : V. Favier compared the accumulation data from the S1C1 ice-core with surface accumulation data from stake measurements and designed Figure 1b. M. Fily and O. Magand participated in the TASTE-IDEA/VANISH 2006-20 2007 summer campaign, and processed density and radionucleides measurements. S. Preunkert performed the chemical measurements, with support from B. Jourdain. M. Legrand provided insights on aerosol signals and layer counting. B. Minster performed the isotopic measurements. V. Masson-Delmotte designed the study and supervised the analyses of the records. M. Werner performed the simulation of the ECHAM5-wiso model while H. Gallée performed the simulation of the MAR model. S. Goursaud dated the S1C1 ice-core and wrote the manuscript with contributions from all co-authors.

**Acknowledgements.** This work and the PhD thesis of S. Goursaud are funded by the ASUMA project supported by the ANR (Agence Nationale de la Recherche, Project n°: ANR-14-CE01-0001). The S1C1 core was drilled within the International TASTE-IDEA (Trans-Antarctic Scientific Traverse Expeditions - Ice Divide of East Antarctica)/ VANISH (Vulnaribility of ANtarctic Ice SHeet and its atmosphere ) programs supported by the ANR (Agence Nationale de la Recherche, Project n°: 30 ANR-07-VULN-0013), during the 2006/2007 summer campaign, with logistical support from Institut Polaire Français-Paul Emile Victor (IPEV). We thank all the field team members who contributed to the success of the 2006-2007 TASTE-IDEA/VANISH summer campaign. GLACIOCLIM-SAMBA observatory is supported by IPEV and INSU. Finally, this work was first presented as a poster to the 2016 IPICS (International Partnership in Ice Core Sciences) conference in Hobart, Tasmania. We are extremely grateful to the scientific and local organization committees for such a stimulating meeting!




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





**Tables**

**Table 1. Statistical analyses of the ECHAM5-wiso outputs: precipitation in cm y$^{-1}$, δ$^{18}$O in ‰ and the 2-meter temperature (2m-T) in °C over the 1979-2007 period.**

| | | Precipitation (cm w.e. y$^{-1}$) | | | | 2m-T (°C) | | | | δ$^{18}$O (‰) | | | |
|---|---|---|---|---|---|---|---|---|---|---|---|---|---|
| | | Min | Max | μ | σ | Min | Max | μ | σ | Min | Max | μ | σ |
| S1C1 | 1947-2007 | 9.7 | 38.0 | 22.0 | 6.9 | | | | | -23.6 | -15.9 | -18.9 | 1.7 |
| | 1979-2007 | 9.7 | 61.3 | 23.1 | 9.8 | | | | | -23.6 | -15.4 | -18.6 | 1.8 |
| READER | 1956-2007 | | | | | -16.1 | -8.6 | -10.8 | 1 | | | | |
| | 1979-2007 | | | | | -11.9 | -8.6 | -10.6 | 0.7 | | | | |
| ECHAM | 1979-2007 | 52.1 | 96.6 | 68.4 | 11.9 | -18.1 | -14.8 | -16.9 | 0.8 | -22.4 | -19.6 | -20.8 | 0.6 |





**Table 2. Linear relationship between the S1C1 $\delta^{18}$O and regional climate records showing the coefficient correlation and the p-value.**

| Period | Resolution | Record | r | p |
|---|---|---|---|---|
| 1997-2007 | annual | S1C1 accumulation | 0.85 | 4.0E-3 |
| 1956-2007 | annual | READER surface temperature | 0.22 | 0.1 |
| 1956-2007 | decadal | READER surface temperature | 0.72 | 2.0E-09 |
| 1979-2007 | annual | READER surface temperature | 0.44 | 2.0E-2 |
| | winter | READER surface temperature | 0.43 | 2.0E-2 |
| | summer | READER surface temperature | 0.21 | 0.5 |
| 1979-2007 | annual | sea ice concentration in Adélie Land | 0.03 | 0.9 |
| 2000-2007 | annual | sea ice concentration in Adélie Land | 0.66 | 8.0E-2 |
| 1950-2007 | annual | wind speed | 0.15 | 0.3 |
| 1979-2001 | annual | wind speed | 0.43 | 4.0E-2 |





**Table 3. Linear relationships between temperature and $\delta^{18}O$ showing the slope, the coefficient correlation and the p-value. The indices associated with "S1C1" indicate the number of the dating.**

| Period | Resolution | Temperature | $\delta^{18}O$ | Slope | r | p |
|---|---|---|---|---|---|---|
| 1956-2007 | annual | reader database | $S1C1_1$ | 0.36 | 0.22 | 0.1 |
| | | reader database | $S1C1_2$ | 0.40 | 0.24 | 8.8E-2 |
| | decadal | reader database | $S1C1_1$ | 1.49 | 0.72 | 2.6E-9 |
| | | reader database | $S1C1_2$ | 1.33 | 0.68 | 5.0E-8 |
| 1979-2007 | annual | reader database | $S1C1_1$ | 1.14 | 0.44 | 2.0E-2 |
| | | reader database | $S1C1_2$ | 1.24 | 0.49 | 1.0E-2 |
| | decadal | reader database | $S1C1_1$ | 1.68 | 0.70 | 2.2E-5 |
| | | reader database | $S1C1_2$ | 1.31 | 0.60 | 5.3E-4 |
| | annual | echam | echam | 0.32 | 0.42 | 2.2E-2 |
| | annual | reader database | echam | 0.43 | 0.48 | 8.8E-3 |
| | annual | echam | $S1C1_1$ | 0.87 | 0.40 | 3.3E-2 |
| | annual | echam | $S1C1_2$ | 1.07 | 0.50 | 5.7E-3 |





**Figures.**

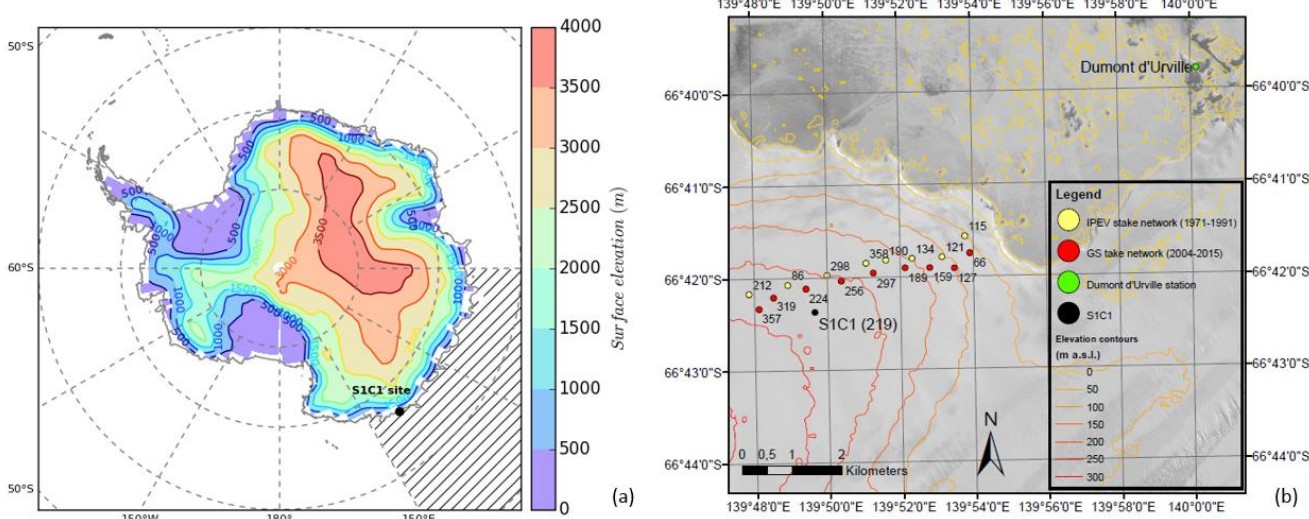

**Figure 1: Maps showing the areas of interest with (a) a map of Antarctica representing the local topography (contours), the location of the S1C1 drilling site and the area used to extract regional sea-ice extent from the Nimbus-7 SMMR and DMSP SSM/I-SSMIS Passive Microwave Data, and (b) an orientation map showing stake networks around S1C1 drilling site. Data from 1971 to 1991 were measurement by IPEV, whereas data from 2004 to 2015 were measured in the framework of GLACIOCLIM-SAMBA (GS) observatory. Elevation lines are from the digital elevation model of Korona et al. (2009), which was computed from SPOT5 images obtained during the fourth International Polar Year (2007–2009). The SPOT5 image is also included as a background map. Labels are mean accumulation values at each stake computed over the period with measurements.**



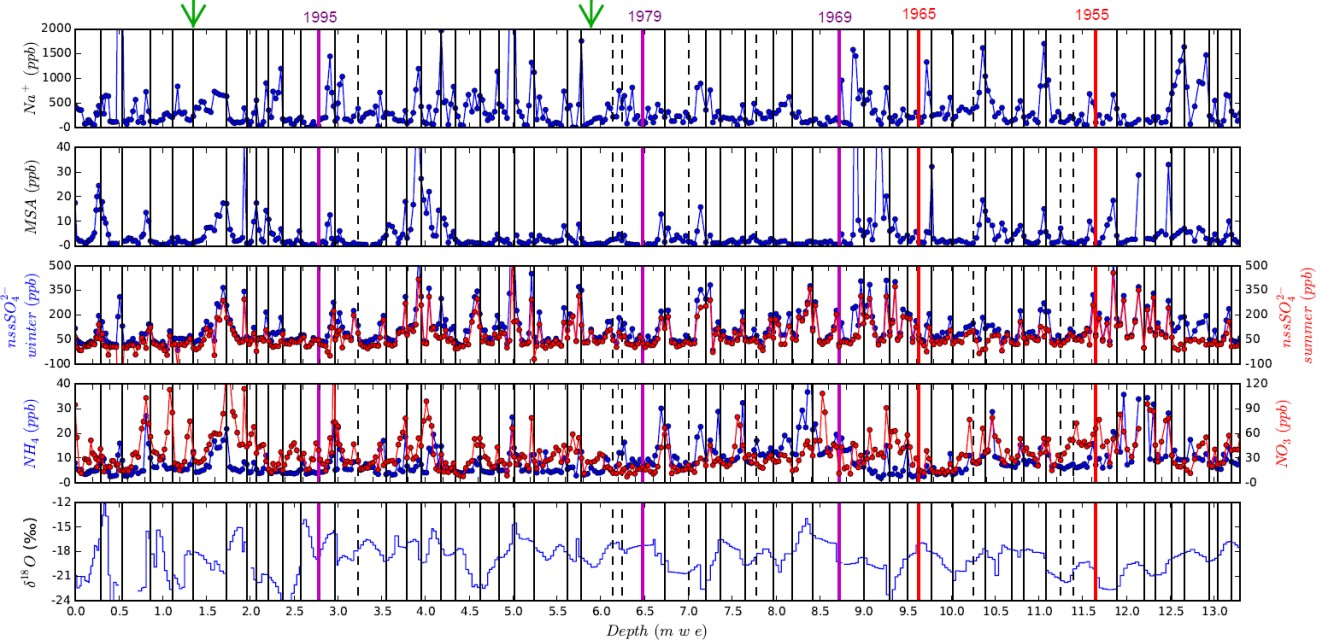

**Figure 2: Dating of the S1C1 core based on sodium Na⁺, dimethylsulfure (DMS), non-sea-salt sulfate (nssSO₄²⁻), ammonium NH₄⁺, nitrate (NO₃), and δ¹⁸O. The black lines correspond to the December-January period of each year (solid lines are the layers for which all sources of information are consistent, while dashed lines are the layers for which we have contrasted results from one or more sources of information), the purple lines to the non-full sea-ice retreat horizons and the red lines to the nuclear test horizons. δ¹⁸O data are shown as horizontal steps because one measurement corresponds to a value averaged over a 5 cm sample. Green arrows stress the years where changes have been implementing for the second dating (see text for details).**





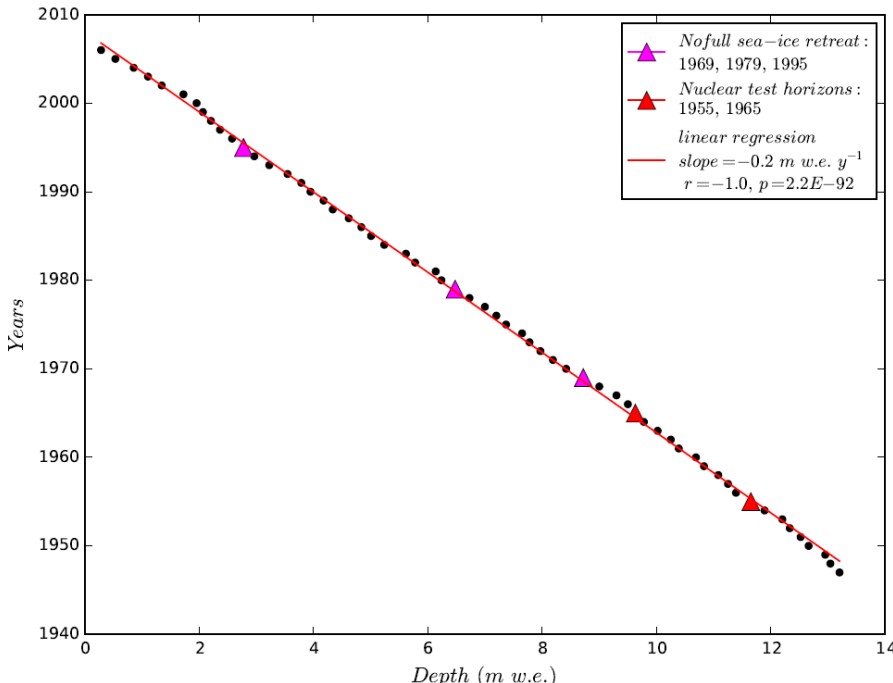

**Figure 3: Depth-age relationship based on the dating of the S1C1 core.**





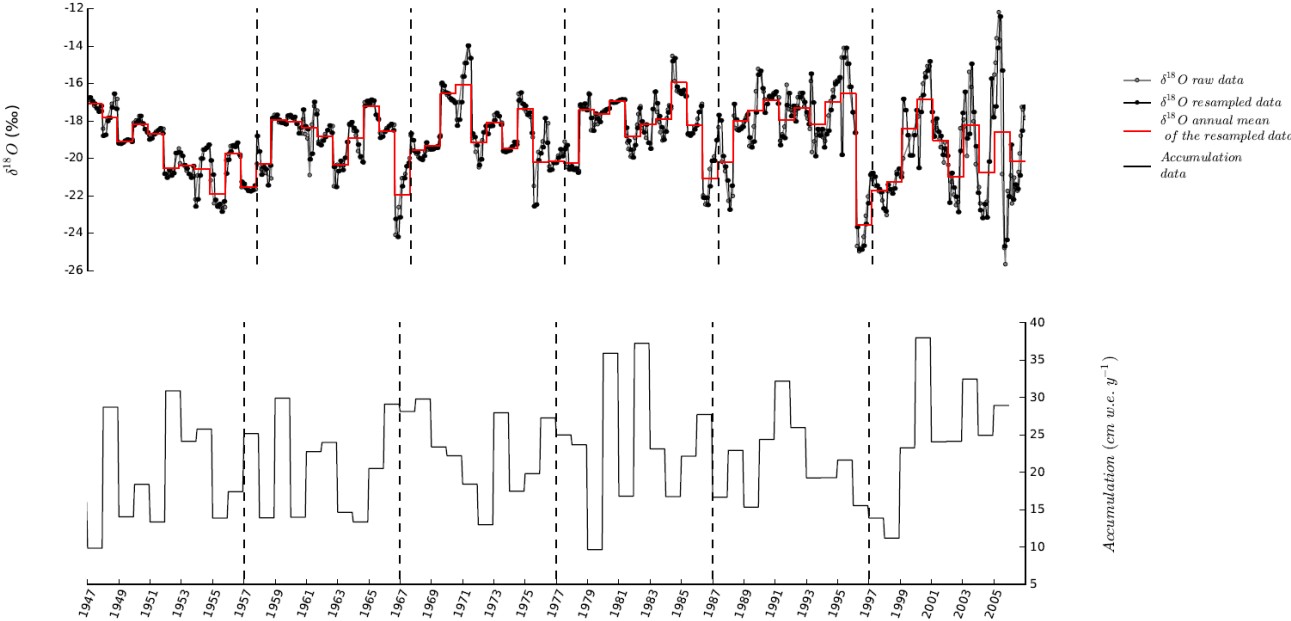

**Figure 4: Time series of δ¹⁸O (in ‰) and accumulation (in cm w.e. y⁻¹) extracted from the S1C1 core. Dark green points correspond to the raw data, while pale green points correspond to the resampled data. Finally, the red and black lines correspond to the annual mean of the resampled data and the accumulation data respectively.**



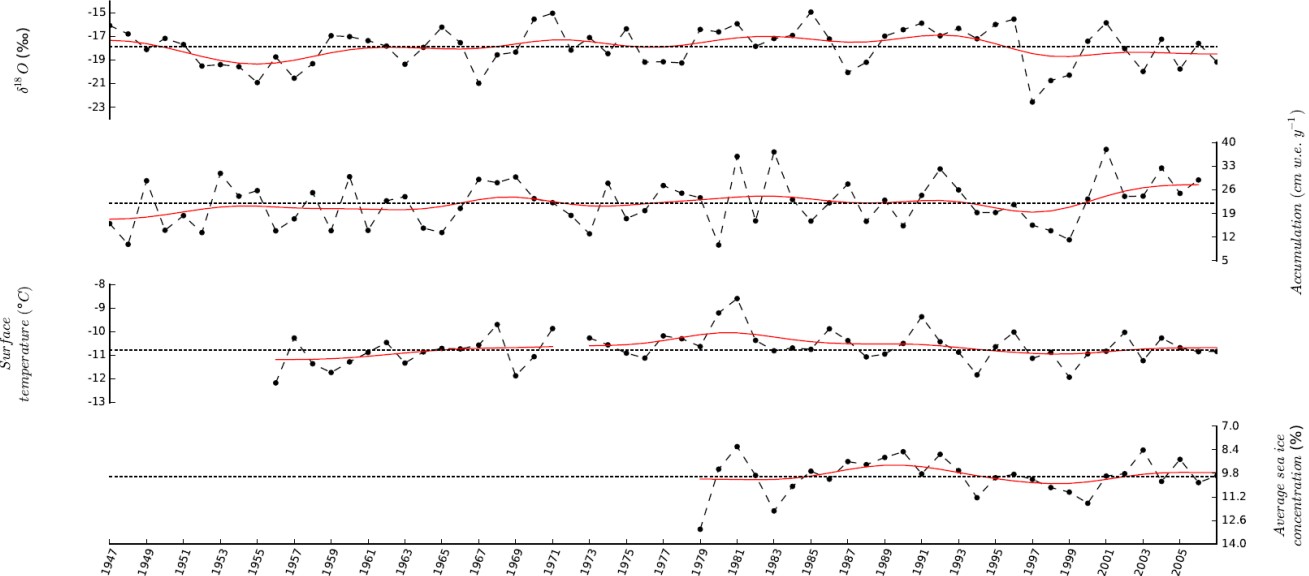

**Figure 5: Comparison of the (resampled) δ18O (‰) and the annual accumulation (cm w.e. y-1) from the S1C1 core, with the near surface temperature (in °C) extracted from the READER data base for Dumont d'Urville Station (https://legacy.bas.ac.uk/met/READER/) and the Average sea ice concentration extracted between 90°E and 150°E from the Nimbus-7 SMMR and DMSP SSM/I-SSMIS Passive Microwave Data (http://nsidc.org/data/nsidc-0051). Note that the y-axis of the lowest panel is reversed, to ensure visual coherency between lower sea-ice extent and higher temperatures.**





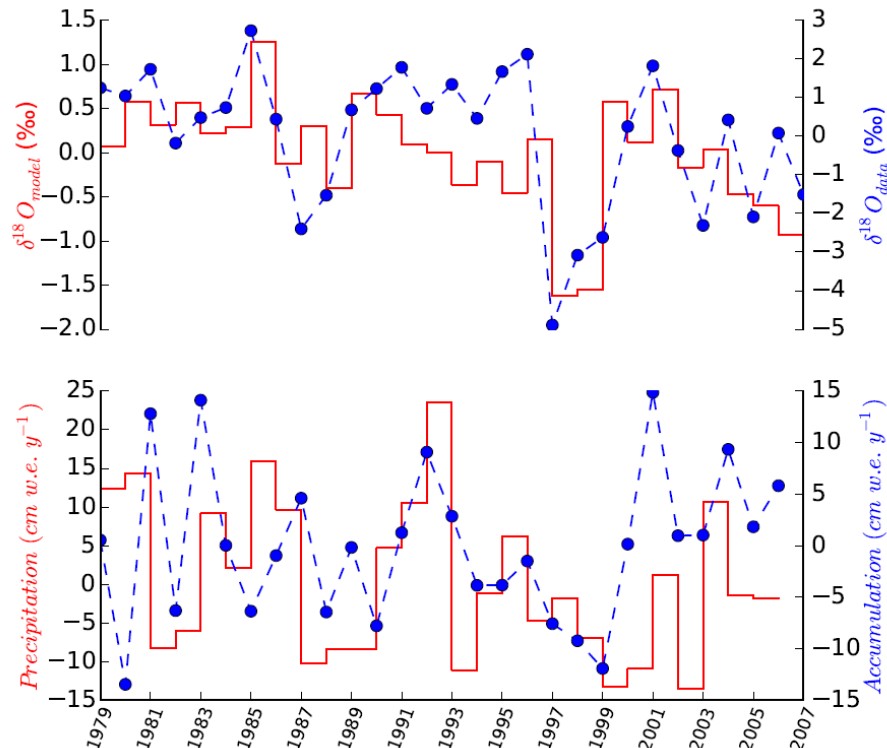

**Figure 6: Model-data comparison of the anomalies (annual value minus the 1979-2007 average value) of δ¹⁸O (‰) and accumulation (cm w.e. y-1). The blue points and dashed lines correspond to the data from the S1C1 core while the red stair steps correspond to the simulations.**



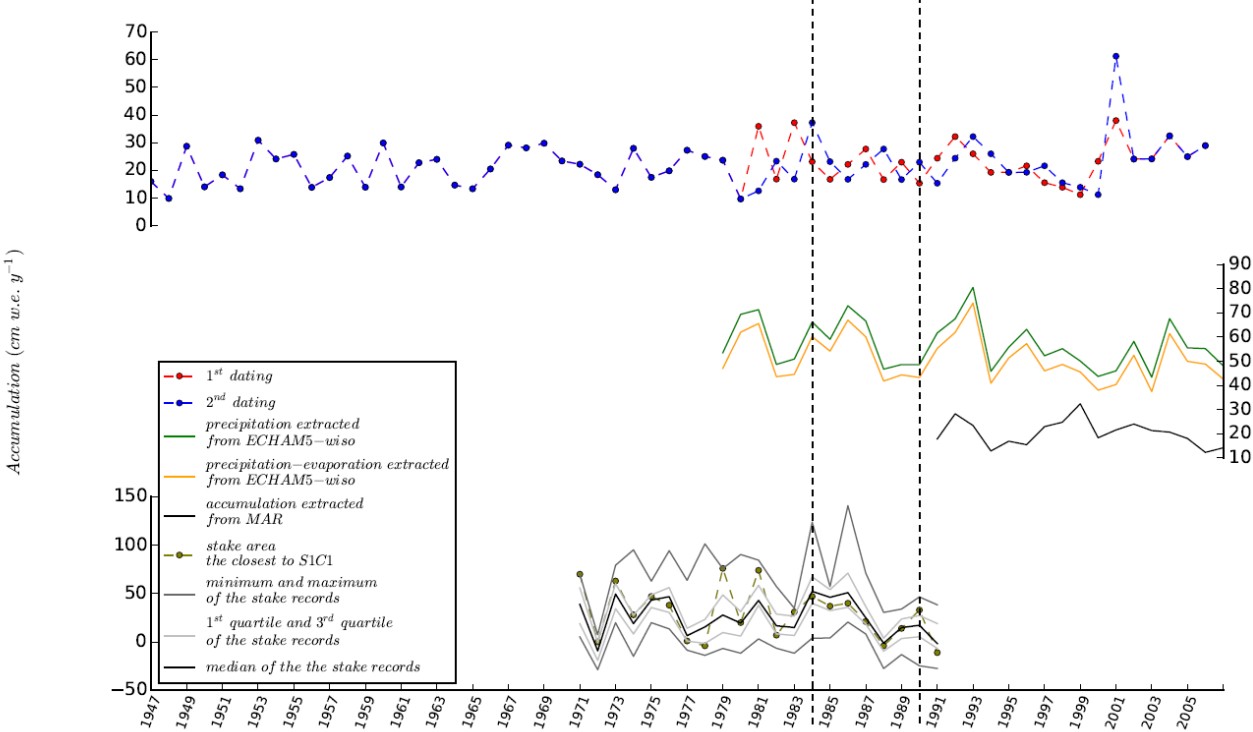

**Figure 7: Comparison of accumulation data. Top panel: S1C1 reconstruction obtained from the first (red) and second (blue) dating. Note that the accumulation resulting from first dating and second dating overlap over 1947-1980 and 2002-2007 so only one colour line appears. Middle panel: precipitation (green) and precipitation minus evaporation (orange) simulated by ECHAM5-wiso (1979-2007), and accumulation simulated by MAR (1991-2007) (black). Lower panel: distribution of accumulation measurements from stake records, illustrated by minima and maxima (grey lines), first and third quartiles (light grey lines), median value of stake measurements (black line) and measurements from one single stake, closest to the S1C1 site (dashed grey line). Two dashed vertical lines correspond to years 1984 and 1990.**





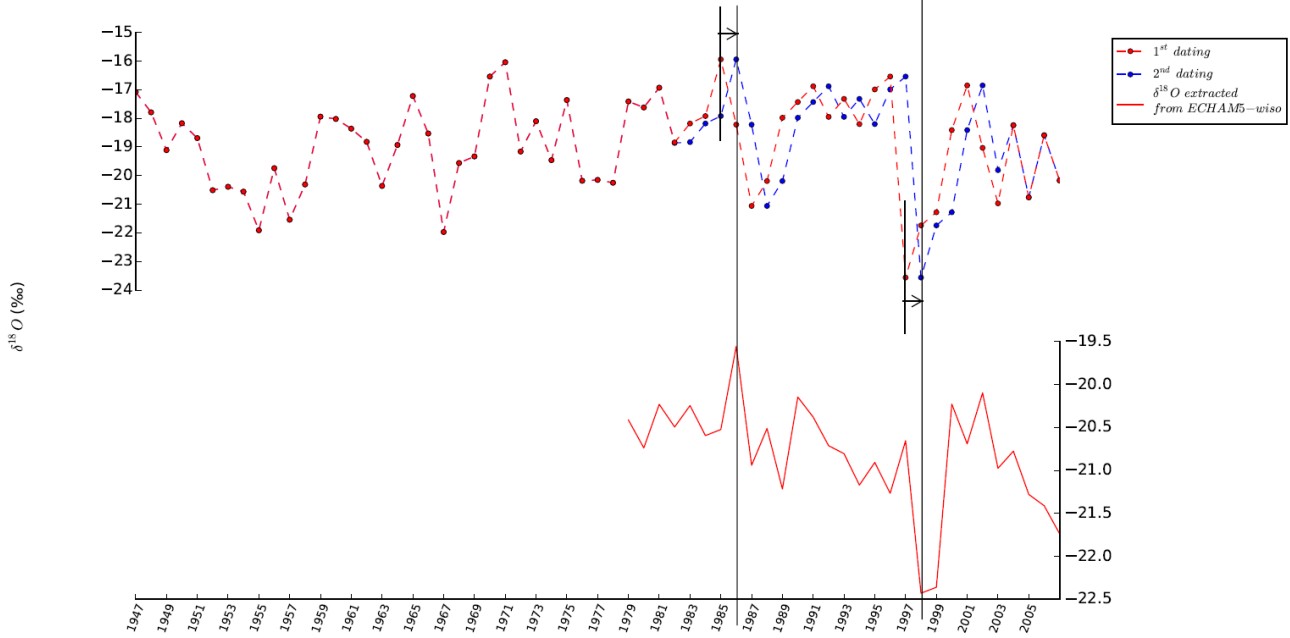

**Figure 8: Comparison of dO18 data. Top panel: annual mean dO18 data from the S1C1 core on the first (red) and second (blue) dating, over 1947-2007. Note that the S1C1 dO18 resulting from first dating and second dating overlap in 1947-1980 and 2002-2007, when only one colour line is displayed. Bottom: annual mean (precipitation-weighted) precipitation dO18 simulated by the ECHAM5wiso-model. Small vertical black lines and arrows illustrate the shift between the two S1C1 age scales for the years with the highest and lowest dO18 values of our record. Long vertical lines illustrate the match between the ECHAM5-wiso and S1C1 peaks used to build the second chronology.**



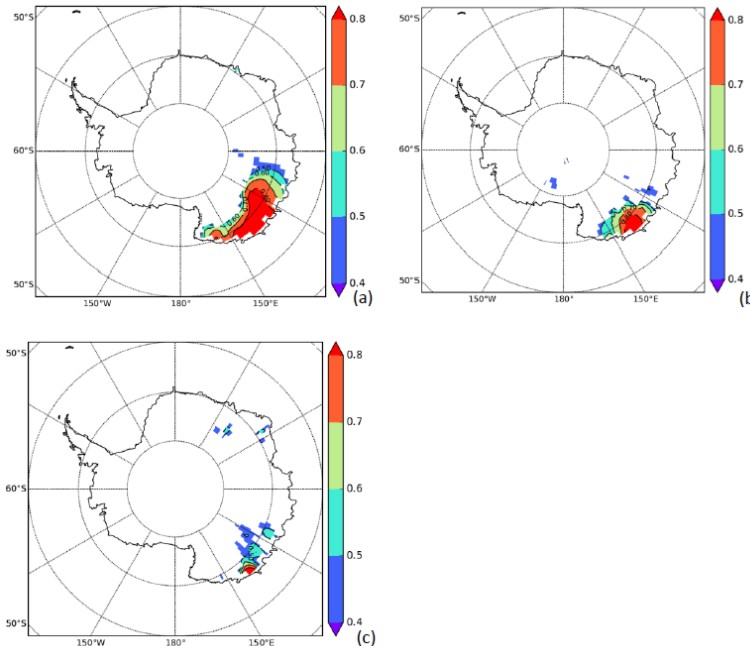

**Figure 9:** **Correlation between simulated annual mean 2m-temperature (a), annual precipitation amount (b), precipitation annual mean δ¹⁸O (c) at the S1C1 site with the same variable at each grid point of the ECHAM5-wiso model for r>0.4 and p<0.03. Only significant values are represented (p<0.05).**