# Peer review of "A sixty year ice-core record of regional climate from Adélie Land, coastal Antarctica"

_The Cryosphere, 2016_

## Author Comment (AC1) · 11 Sep 2016

The affiliation of Dr Martin Werner is wrong in the manuscript. It should have been written instead: Alfred Wegener Institute for Polar and Marine Research (AWI), Bremerhaven, Germany

---

## Referee Comment (RC1) · E. Isaksson (Referee) · 29 Oct 2016

The paper present and discuss the data from a 22 m deep ice core covering 60 years drilled from a coastal site in Adelie Land, an area well known for the strong katabatic winds. The ice core data are discussed in relation to outputs from a high resolution atmospheric general circulation model including water isotopes. Because of the specific meteorological conditions with strong winds the dating of this ice core has been a big challenge- and that is one reason for the lack of such data from this particular region. However, because of the proximity to Dumont d'Urville, various other glaciological and meteorology data has been collected here for several decades. Thus, there are some valuable scientific background data that are helpful for both dating and interpretation of this ice core. A major part of the paper focuses on the dating of the core- something that is important for this particular site for the reasons just mentioned. The authors do

a thorough job describing the procedure in great details, and verifying the traditional dating with model output. I find this to be an important scientific contribution because dating is something that is usually regarded as a method and thus do not receive too much attention. The main conclusion in the paper is that one core can indeed accurately capture major annual anomalies and multi-decadal variations. During recent years there has been lots more focus on ice core data from coastal regions in Antarctica and therefore the results from the drill site presented here is highly interesting. It is important to collect field data also from "difficult" areas for validation of models so I feel that this paper is a very important contribution, and also an inspiration for such work. Although the paper is presenting valuable data there are a few issues to consider before it is accepted.

1. It is very important to have field data for validation of models but it is also crucial that the field data are of good quality. There are a number of pieces of information missing in the manuscript in order for the reader to make this judgement. Not much data on the general glaciology and meteorology are provided- and the available information is not collected in one place which makes it harder for the reader. It could be good to collect the information that exists in a separate chapter - "area description"-in the beginning of the paper. For instance, the wind direction and wind speed information (chapter 5.3.1.) should be appearing in the general introduction because this is such fundamental information for all interpretation.

2. Information about spatial distribution of accumulation is missing in the paper. A single ice core and its regional representativeness is always an issue for discussion. A first step is to look at the snow layers using GPR (see recommendations in Eisen et al. 2008). In chapter 2.1 GPR measurements are briefly mentioned, and seem to have been collected at the time of the core drilling. Then one might wonder why that is not included here? When this core in addition comes from an areas so affected by katabatic winds this definitely has to be paid more attention to. Also, I think it is necessary to include some more information about the stake data for the same reason. As a reader

I am not entirely convinced that the good agreement with major annual anomalies and multi-decadal variations is not pure coincidence. . . .. . .

3. There is no information about ice layers and potential effect of melting.

4. The paper is generally heavy to read and it looks like the text is based on a thesis with different chapters. I suggest the authors try to re-organize- and maybe also shorten it a bit. Dating is a main aspect of this paper as it is written now so maybe that should be reflected in the title?

Tables and figures

Table. 2. Specify % for r and p

Table 3. Use capital letters for READER and ECHAM as in other tables.

Specify % for r and p, unit for slope

Figure 2. Spelling of DMS is not English

Figure 3. Different colors are needed- hard to distinguish between the ones chosen.

Figure 7. Poor color choice also for this figure. Add labels and units for y-axes. Very difficult to distinguish the different information from one another in the lower panel.

Figure 8. This figure would benefit from being re-drawn. Labels should be better placed. Legends need to be bigger. All the "d18O" in the Figure caption should be changed to the right format.

Minor issues

There are a number of language problems, typos etc. for the authors to take care of. Here is a small selection- but there are many more.

p. 5 line 23. A slight rewrite to remove at least one of the three "cleaned"

p. 5 line 27. Remove a single "e"

p. 9 line 4-9. There are many very long sentences in the paper that makes it hard to read. Here is one of many examples.

p. 10 line 5. long term. . . ? Uncomplete sentence

p. 11 line 5. Should be "links"

p. 11 line 32. Should be "the period"

p. 12 line 8. Should be cha

---

## Referee Comment (RC2) · B. Stenni (Referee) · 4 Nov 2016

The paper by Goursaud and co-authors is presenting a new isotopic and snow accumulation rate records obtained from an ice core drilled in the Adelie Land coastal area (in the proximity of the French Dumont d'Urville station) in a site which is characterized by a relatively high snow accumulation rate (about 220 mm w.eq. yr-1) and covering the 1947-2006 period. The paper is much focused on the dating issues and on the comparison with the data simulated from the high resolution atmospheric general circulation ECHAM5-wiso model ($\delta$18O as well as precipitation) and with the regional atmospheric model MAR. The authors also suggest a method to improve the dating considering particular isotopic signature in the inter annual variability of ECHAM5-wiso isotopic data. The obtained results suggest that also a single ice core from a coastal area can capture the main climatic signals although a multi-core approach should be

desirable in order to reduce the stratigraphic noise which is unavoidable. The paper is interesting, the data are well presented and I have found the reading quite smooth although some parts could be reduced a little (e.g. paragraph 3.1.3). I recommend its publication after the authors have been considered to the following comments.

Page 2, line 3: ice core chronology: the period covered by the core should be 1947-2006 rather than 2007, in fact the drilling has been carried out in January 2007, so the topmost snow layers should at maximum be referred to the year 2006. Please, check this in the whole manuscript, as well as in the tables, figures and related captions. I did not understand if this is just a refuse or a real mistake. Page 3, line 3: the reference Ahmed is not correct! Please change into "PAGES 2k Consortium, 2013". Page 3, line 5: the reference Jones et al., in press. . ... is published. Page 3, line 26: please add here Schlosser et al., 2008 (Neumayer station data) as well as Stenni et al., 2016 (TC). Page 4, line 6: delete the "." Page 4, lines 25-27: the sentence "Data obtained . . ... .. context." is not clear at all. Please explain what do you mean. Page 4, line 31: please change the Laboratory of Glaciology into LGGE, already defined before. Page 5, lines 13-14: the citations Delmas and Pourchet (1977) and Magand et al (??) (2009) are lacking in the Reference. Page 5, line 21: why not shown? May you consider to have a figure on this in the Supplement? Page 5, line 27: delete "e" and add "the". The method is the "$CO_2/H_2O$ equilibration method". Please add. Page 5, line 28: add $\pm$ before 0.05 and after ‰ add (1 sigma). Page 5, line 29: I would not refer to the figure 2 here but rather in the result section. Page 6, line 10: delete "in" after Antarctic sites and add "the". Page 6, line 29: add "from" before the Nimbus Page 7, line 25: the citation Bintanja (2000) is lacking in the Reference. Page 7, lines 27-29, line 31: the citations Kessler (1969), Lin et al. (1983), Meyers et al. (1992), Levkov et al. (1992), Morcrette (2002) and Galle et al. (2013) are lacking in the Reference. Page 8, line 4: the authors refer here to one year DDU record of precipitation. Are these data available? Are these the same data cited later on in the text (paragraph 3.3.1)? Please, explain better. Perhaps the data could be added in the Supplement? Page 9, line 1: paragraph 3.1.3: I suggest to reduce a little this paragraph by using a table or a figure. . ... Page 10,

line 20: add a space between "calculated" and " from". Line 10, line 28: less depleted values .... Different seasonal pattern: the mean $\delta$18O at S1C1 is -18.9‰ not very far from -18 at DDU considering the difference in elevation. I mean the results seems consistent, obviously less depleted than DC........ Then regarding the difference with Dome C it's not surprising but I cannot comment on the different seasonal pattern if all the data (DDU) are not provided as a figure .... Moreover, considering the large inter-annual variability of Antarctic climate and the only less than 1-year record at DDU I do not think that it is appropriate to discuss about different seasonal patterns. Page 10, line 29: Stenni et al., 2016, now published (to be changed also in the Reference). Page 11, line 9: when comparing S1C1 site with Law Dome you have to consider the extremely different snow accumulation rate between the two sites. Please add something about this. Also add the Johnsen (1977) citation about post-depositional effects. Page 11, 13-22: please refer to the appropriate figures, 4 and 5, in the text otherwise it is not easy to follow. Page 11, line 23: delete "accumulation" after $\delta$18O or an "and" is lacking? Please, correct. Page 11, line 33: at Dome C we used daily values or monthly values and obtained a slope 0.49‰°C. Not sure what you have used here, please explain. Page 12, line 1: please check the reported statistical values. Also there seem to be a typo error .... Page 12, line 2-3: at Dome C we considered daily values, so we did not exclude the seasonal cycle. We also obtained a higher slope (1.4) if considering the only 3 annual values, but its significance is low since it is calculated on 3 years only. Here, on the other hand, you are considering inter-annual variability. Page 12, line 14: rather than metamorphism I would say exchanges between surface snow and water vapour. Page 12, line 26: the first reported values (r=-0.48 and p=8.0E-3) seem significant. Please, check. Page 13, line 21-22: in Antarctica this problem was nicely shown by Frezzotti et al, 2007 JGR. Also for snow accumulation rate values as those found by the authors for this Adelie Land site S1C1 and also considering the wind effects (see comment by E. Isaksson)! Page 13, line 27: add "scale" after " horizontal". Page 14, line 15: the year should be 1986 and not 1985. Page 15, line 7: add a space between "the" and "period". Page 15, line 10: what do you mean by

"climatic deposition signal" ... not clear. Page 16, line 3: why "his"? Page 16, line 6-7: comparison with the ECHAM5-wiso output BUT with which chronology? Page 16, line 9: contradictory: may you explain better? Page 16, line 20: Stenni et al., 2016. Page 16, line 21: see also the conclusions by Schlosser et al., 2016 (ACP) about the inter-annual difference. Page 16, line 28: if I am not wrong, this statement "but a significant and weaker correlation between $\delta$18O from the S1C1 core and simulated by ECHAM5-wiso" depends on the age scale considered, isn't it? Page 17, line 10: this adjective "coherent" seems in contradiction with what you have just said few lines before. Page 18, line 4: change sake into stake Page 18, line 7: please consider that the difference in the snow accumulation rate (and wind action) at the different sites and their effects on the diffusion effects and so on the smaller amplitude of the seasonal isotopic cycle. See comment before. Page 18, line 10-12: the sentence " We stress .... relationship." seems not valid if I consider $\delta$ core and Techam. May you check? Page 18, line 17: I would also add that precipitation sampling at DDU would be desirable. Page 18, line 34: the reference Ahmed is not correct! Please change into "PAGES 2k Consortium, 2013". And also Jones et al. is published. Page 19, line 5: please specify what is PSA2. References: some are not completed (Jones et al, Lemeur submitted, Masson-Delmotte et al., 2008 the authors list is not completed) and some must be changed from discussion to final accepted papers (Ritter, Schlosser). Please, make a careful check. Table 1. The precipitation column is not only precipitation (ok for ECHAM) but in the case of the core data is accumulation. What is u? I suppose average.... Table 2: be careful about the year 2007 (the period covered by the core )...... see my comment above (page 2, line 3). Table 3: something is lacking in the first column: You should have 1956-2006 (annual and decadal) and then the same for 1979-2006. Why you did not discuss in the text the relationship between Techam and $\delta$18O S1C11 and S1C12, if I am not wrong? (see my comment above Page 18, line 10-12). Figure 1: add "hatched area" in the caption after "Passive Microwave Data". Figure 2: in the caption: you do not have dimethylsulfure data but MSA data!! I would also make the labels larger than they are. Figure 4: the different lines are not distinguishable. Also in

this case I would use larger labels. In the caption: resampled data: specify with which step. The annual mean is calculated by the annual layer dating? Specify. Figure 5: Also in this case I would use larger labels. In the caption, some typos.... Figure 6: I had some difficulty in comparing the values referring to the red (cityscape) and the blue lines. Moreover, check also here the correct period covered by the core. The first year is 2006, the accumulation record seems OK but for the $\delta$18O why you have a value for 2007? Here the dimensions of the different labels are OK. Figure 7: Also in this case I would use larger labels. Figure 8: Also in this case I would use larger labels. Also here the first $\delta$18O seems to be 2007.... To be checked.

---

## Author Response (AR1)

**Reply to reviewers' comments.**

Dear Editor,

5   We would like to thank the two reviewers for their detailed comments that helped us improve the text, tables and figures of our manuscript. We have taken into account their remarks in the revised version that is provided, together with a point by point answer to comments.

We also would like to stress here that, following one comment from Dr. Stenni, we have identified a mistake in our ice core
10  chronology due to a wrong first year (2007 instead of 2006). We deeply thank Dr. Stenni for identifying this issue and have therefore corrected our initial chronologies, repeated all calculations and corrected results in all figures as well as in the numbers provided in the text. This correction only has minor implications on our key findings.

**Reply to comments from Dr. Elizabeth Isaksson**

The reviewer comments are displayed in italics, followed by our answers.

*The paper present and discuss the data from a 22 m deep ice core covering 60 years drilled from a coastal site in Adélie*
*Land, an area well known for the strong katabatic winds. The ice core data are discussed in relation to outputs from a high*
20  *resolution atmospheric general circulation model including water isotopes. Because of the specific meteorological*
*conditions with strong winds the dating of this ice core has been a big challenge- and that is one reason for the lack of such*
*data from this particular region. However, because of the proximity to Dumont d'Urville, various other glaciological and*
*meteorology data has been collected here for several decades. Thus, there are some valuable scientific background data that*
*are helpful for both dating and interpretation of this ice core.*

We thank Dr. Isaksson for her appreciation of the new data.

*A major part of the paper focuses on the dating of the core- something that is important for this particular site for the*
30  *reasons just mentioned. The authors do a thorough job describing the procedure in great details, and verifying the*
*traditional dating with model output. I find this to be an important scientific contribution because dating is something that is*
*usually regarded as a method and thus do not receive too much attention.*

We share the concern for the importance of chronologies. Please note that the chronology has been partly corrected (see above), without any consequence for the key findings of our analyses.

*The main conclusion in the paper is that one core can indeed accurately capture major annual anomalies and multi-decadal variations. During recent years there has been lots more focus on ice core data from coastal regions in Antarctica and therefore the results from the drill site presented here is highly interesting. It is important to collect field data also from "difficult" areas for validation of models so I feel that this paper is a very important contribution, and also an inspiration for such work. Although the paper is presenting valuable data there are a few issues to consider before it is accepted.*

We again thank Dr. Isaksson and have carefully considered the 4 main points of comments.

*1. It is very important to have field data for validation of models but it is also crucial that the field data are of good quality. There are a number of pieces of information missing in the manuscript in order for the reader to make this judgement. Not much data on the general glaciology and meteorology are provided- and the available information is not collected in one place which makes it harder for the reader. It could be good to collect the information that exists in a separate chapter - "area description"-in the beginning of the paper. For instance, the wind direction and wind speed information (chapter 5.3.1.) should be appearing in the general introduction because this is such fundamental information for all interpretation.*

In order to provide more information about the general meteorological context (wind characteristics), but also to keep the manuscript concise, we have added a brief description of the site characteristics upfront in the introduction. Wind direction and wind speed characteristics at DDU have been explicitly added (cf p. 3 l. 13-16): "These features lead to dominant southerly (160 plus or minus 20°) intense katabatic winds (Wendler et al., 1997). During summer open sea conditions, a reverse pattern of wind direction can occur, associated with sea breezes. At Dumont d'Urville station (hereafter DDU), Périard and Pettré (1993) reported an average wind speed of 10 m s$^{-1}$, as well as events when wind speeds exceed 20 m s$^{-1}$ during several days (with an observed local maximum wind speed at 90 m s$^{-1}$)."

*2. Information about spatial distribution of accumulation is missing in the paper. A single ice core and its regional representativeness is always an issue for discussion. A first step is to look at the snow layers using GPR (see recommendations in Eisen et al. 2008). In chapter 2.1 GPR measurements are briefly mentioned, and seem to have been collected at the time of the core drilling. Then one might wonder why that is not included here? When this core in addition comes from an areas so affected by katabatic winds this definitely has to be paid more attention to. Also, I think it is necessary to include some more information about the stake data for the same reason. As a reader I am not entirely convinced that the good agreement with major annual anomalies and multi-decadal variations is not pure coincidence.*

We agree with Dr. Isaksson that providing information on the variability of accumulation in an area so much exposed to katabatic winds is critical. We therefore expanded the discussion section with respect to information available from stake data, and reported what existing lines for information with respect to the variability of accumulation in the region. (Section 4.3, p.15 l.22-25):

"Indeed, the mean accumulation given by the stake located at D10 is 25.6 cm w.e y$^{-1}$ and the standard deviation of mean accumulation values from the 14 stakes located within the first 10 km is 12.0 cm w.e y$^{-1}$. A high variability reaching tens of milimeter over few meters was also reported by Genthon et al. (2007) and Favier et al. (2011) over an ablation site located 4 km from D10."

We also expanded the description of processes (distribution of the precipitation and wind driven erosion or deposition) affecting the signal in the S1C1 core.

*3. There is no information about ice layers and potential effect of melting.*

In our initial manuscript, we provided information on negative summer temperature at the sampling site during a year that appears to be representative of average summer conditions (-5°C). In response to the request to provide more information, we have again expanded Section 4.3 to explicitly address the issue of potential melting and refreezing. For this purpose, we refer to the analysis of small refreezing ice layers from the stake network of the Glacioclim observatory and report precipitation events recorded at D10 during the Glacioclim campaigns. Based on this evidence, we conclude that liquid precipitation, melting and refreezing processes have very limited impacts in this region (Section 4.3 p.15 l.31- p.16, l.7):

"However, refreezing of precipitation or melting in this region does not seem to be enough significant to be counted as noise. Indeed, 2.5-m snow cores were collected every 0.5 km from the 156-km stake network of Glacioclim observatory. These surface cores were used for density measurements but also for a quick stratigraphy analysis. Small refreezing ice layers exceeding 2 mm can have been identified. It was found that these layers were found the same depth. At D10 and around, only one event of very large refreezing was observed, following a high precipitation event between December, the 31$^{th}$ of 2014 and January, the 01$^{st}$ of 2014. At D10, accumulated thickness of ice layers within the first 2.5 m of ice exceeded 25 cm w.e. and ice layer were sometimes 5 to 6 cm thick. However, this event only occurred once over the last 60 years according to the observations made at the Meteo-France station. In order to confirm this point, over the last 10 years, we only observed one ice layer thicker than 3 mm at D10. Indeed, a 3 cm thick ice layer was observed at 60 cm in the surface core collected in 2008-2009. As a consequence, melting and liquid precipitation occur at D10, but the effect on percolation is limited."

*4. The paper is generally heavy to read and it looks like the text is based on a thesis with different chapters. I suggest the authors try to re-organize- and maybe also shorten it a bit. Dating is a main aspect of this paper as it is written now so maybe that should be reflected in the title?*

We apologize for the fact that the manuscript does not easily read. The first author of the manuscript is a PhD student and she was eager to share with readers the diversity of investigations that she has conducted: the text is not based on a thesis, but the manuscript will be part of a PhD thesis scheduled for 2018. We have done our best to reduce some paragraphs that were not necessary (deleting a few sentences) but have also added information to answer to the comments of the reviewers.

*Tables and figures*

*Table. 2. Specify % for r and p*

The correlation coefficient r and the p-value are dimensionless and are not expressed in %.

*Table 3. Use capital letters for READER and ECHAM as in other tables. Specify % for r and p, unit for slope*

We used capital letters for READER and ECHAM as in other tables and added the unit for the slope.

*Figure 2. Spelling of DMS is not English*

Figure 2. We corrected Dimethyl sulfure (DMS) to Methansulfonic acid (MSA).

*Figure 3. Different colors are needed- hard to distinguish between the ones chosen.*

Figure 3. We changed the colours of the test horizons to make the figure more readable.

*Figure 7. Poor color choice also for this figure. Add labels and units for y-axes. Very difficult to distinguish the different information from one another in the lower panel.*

Figure 7. The y-axis label and its unit are written on the left side of the figure as for other graphics. We hope that the editorial setting will allow this figure to be printed in a larger format so that to enhance readability.

*Figure 8. This figure would benefit from being re-drawn. Labels should be better placed. Legends need to be bigger. All the "d18O" in the Figure caption should be changed to the right format.*

Figure 8. We re-drew the Figure using cityscape lines for all the plots. As for the previous figure, the y-axis label is the same for all the plots and therefore, has been placed to the right side of the Figure. We made the legends bigger.

*Minor issues*

*There are a number of language problems, typos etc. for the authors to take care of. Here is a small selection - but there are many more.*

*p. 5 line 23. A slight rewrite to remove at least one of the three "cleaned"*

*p. 5 line 27. Remove a single "e"*

*p. 9 line 4-9. There are many very long sentences in the paper that makes it hard to read. Here is one of many examples.*

*p. 10 line 5. long term. . . ? Uncomplete sentence*

*p. 11 line 5. Should be "links"*

5   *p. 11 line 32. Should be "the period"*

*p. 12 line 8. Should be cha*

We corrected all these minor issues (omission or orthography mistakes) and shortened sentences that were difficult to read.

**Reply to comments from Barbara Stenni**

*The paper by Goursaud and co-authors is presenting a new isotopic and snow accumulation rate records obtained from an ice core drilled in the Adelie Land coastal area (in the proximity of the French Dumont d'Urville station) in a site which is*
15   *characterized by a relatively high snow accumulation rate (about 220 mm w.eq. yr-1) and covering the 1947-2006 period. The paper is much focused on the dating issues and on the comparison with the data simulated from the high resolution atmospheric general circulation ECHAM5-wiso model (δ18O as well as precipitation) and with the regional atmospheric model MAR. The authors also suggest a method to improve the dating considering particular isotopic signature in the inter-annual variability of ECHAM5-wiso isotopic data. The obtained results suggest that also a single ice core from a coastal*
20   *area can capture the main climatic signals although a multi-core approach should be desirable in order to reduce the stratigraphic noise which is unavoidable. The paper is interesting, the data are well presented and I have found the reading quite smooth although some parts could be reduced a little (e.g. paragraph 3.1.3). I recommend its publication after the authors have been considered to the following comments.*

25   We thank Dr B. Stenni for her interest and her recommendations.

*Page 2, line 3: ice core chronology: the period covered by the core should be 1947- 2006 rather than 2007, in fact the drilling has been carried out in January 2007, so the topmost snow layers should at maximum be referred to the year 2006. Please, check this in the whole manuscript, as well as in the tables, figures and related captions. I did not understand if this*
30   *is just a refuse or a real mistake.*

We thank Dr B. Stenni for having directed us toward a dating mistake. Indeed, we had made a mistake when copying/pasting the resampled $\delta^{18}O$ and had shifted all to one year. So the initial S1C1 chronology was dated from 1947 to 2007 instead of 1946-2006. Therefore, in the following of the study, the selected period for all the climatic records was also the period 1947-

2007. We have corrected our data (cf. Supplementary Material), and re-processed all the statistical calculations. The main changes are the lack of significant correlation analysis results between $\delta^{18}O$ from the S1C1 core (1$^{st}$ dating) and the DDU near surface temperature at the annual scale, and lack of significant correlation analysis results between $\delta^{18}O$ from the S1C1 core (1$^{st}$ dating) and the SAM or the Niño index. Changes were made in the body of the paper, in the tables (Tables 2 and 3) and in the Supplementary Material. We also revised the alternative dating, based on the comparison between $\delta^{18}O$ from the S1C1 core and $\delta^{18}O$ outputs simulated by ECHAM5-wiso Instead of one-year shift, this alternative dating had two-year shift compared to the first one. Details from the choice of the added/deleted identified annual layer is detailed in Section 4.1.

*Page 3, line 3: the reference Ahmed is not correct! Please change into "PAGES 2k Consortium, 2013". Page 3, line 5: the reference Jones et al., in press. ... is published.*
Done

*Page 3, line 26: please add here Schlosser et al., 2008 (Neumayer station data) as well as Stenni et al., 2016 (TC).*
Done

*Page 4, line 6: delete the "."*
Done

*Page 4, lines 25-27: the sentence "Data obtained. ... context." is not clear at all. Please explain what do you mean.*
To make it clear, the sentence was rewritten (p.5, l.3-6):
"Data obtained during 3 days at 25 cm depth show a mean value of -5.0 °C. At DDU and Dome C, July 2007 temperatures were close to the climatological average conditions from the period 1995-2015. As a result, our punctual measurements from summer 2007 are assumed to be representative of average summer conditions."

*Page 4, line 31: please change the Laboratory of Glaciology into LGGE, already defined before.*
Done

*Page 5, lines 13-14: the citations Delmas and Pourchet (1977) and Magand et al (??) (2009) are lacking in the Reference.*
Done

*Page 5, line 21: why not shown? May you consider to have a figure on this in the Supplement?*
Both 1955±1 AD and 1965±1 AD peaks are shown in Figure 2. To make this more explicit, Figure 2 is now called at the end of the sentence (red vertical lines, Fig. 2).

*Page 5, line 27: delete "e" and add "the". The method is the "CO2/H2O equilibration method". Please add.*

The "$CO_2/H_2O$ equilibration method" was detailed for the $\delta^{18}O$ measurements.

*Page 5, line 28: add ± before 0.05 and after ‰ add (1 sigma).*

5   Done

*Page 5, line 29: I would not refer to the figure 2 here but rather in the result section.*

Following the resolution of the $\delta^{18}O$ measurements, instead of referring to Figure 2, we now refer to Table S1 from the Supplementary Material (p.6, l.4-5):

10   "The accuracy of each measurement is ±0.05 ‰ ($1\sigma$) (Table S1 Supplementary Material)."

*Page 6, line 10: delete "in" after Antarctic sites and add "the".*

Done

15   *Page 6, line 29: add "from" before the Nimbus Page 7, line 25: the citation Bintanja (2000) is lacking in the Reference.*

Done

*Page 7, lines 27-29, line 31: the citations Kessler (1969), Lin et al. (1983), Meyers et al. (1992), Levkov et al. (1992), Morcrette (2002) and Galle et al. (2013) are lacking in the Reference.*

20   Done

*Page 8, line 4: the authors refer here to one year DDU record of precipitation. Are these data available? Are these the same data cited later on in the text (paragraph 3.3.1)? Please, explain better. Perhaps the data could be added in the Supplement?*

The only $\delta^{18}O$ measurements at DDU were those communicated by Jean Jouzel. Not to confuse the reader, we deleted the

25   sentence here and presented these data only in Section 3.3.1 (p.10, l. 22-25), as previously written.

*Page 9, line 1: paragraph 3.1.3: I suggest to reduce a little this paragraph by using a table or a figure. ...*

As suggested, the new Table 1 details the ooccurrence of species not showing a summer peak for an identified summer (data missing in summer, peaks not identified at all, or 2 or 3 peaks appearing for the same year). As a result, the text paragraph

30   has been shortened.

*Page 10, line 20: add a space between "calculated" and "from".*

Done

*Line 10, line 28: less depleted values . ... Different seasonal pattern: the mean δ18O at S1C1 is -18.9‰ not very far from -18 at DDU considering the difference in elevation. I mean the results seems consistent, obviously less depleted than DC. ...*
*Then regarding the difference with Dome C it's not surprising but I cannot comment on the different seasonal pattern if all the data (DDU) are not provided as a figure ...*

5  In our discussion of seasonal patterns, we referred to the timing of the maxima, which are reported in the text for DDU and Dome C: "the highest values in March (-10.0 ‰) and November (-11.5 ‰)" (p.10, l.25), and "…with maximum values in December-January" for Dome C (p.10, l.31). We agree that the comparison between the few data points from DDU and S1C1 is not conclusive (quite similar mean levels, and impossibility to conclude about the average seasonal cycle from just one year of sampling at DDU).

*Moreover, considering the large inter-annual variability of Antarctic climate and the only less than 1-year record at DDU I do not think that it is appropriate to discuss about different seasonal patterns.*
We have been very explicit about the limits to such comparisons:" This comparison is of course limited by the short dataset
15  available for DDU and the fact that these measurements were not performed for the same time period."

*Page 10, line 29: Stenni et al., 2016, now published (to be changed also in the Reference).*
Done

20  *Page 11, line 9: when comparing S1C1 site with Law Dome you have to consider the extremely different snow accumulation rate between the two sites. Please add something about this.*
We have added explicitly a comparison of differences in accumulation rates between the S1C1 site and Law Dome (p.11, l.11-12):
"Note that the annual accumulation rate at Law Dome (64 cm. w.e. $y^{-1}$) is ~3 times higher than in the S1C1 core (21.9 cm.
25  w.e. $y^{-1}$)."

*Also add the Johnsen (1977) citation about post-depositional effects.*
Done

30  *Page 11, 13-22: please refer to the appropriate figures, 4 and 5, in the text otherwise it is not easy to follow.*
Done

*Page 11, line 23: delete "accumulation" after $δ^{18}O$ or an "and" is lacking? Please, correct.*
Done

*Page 11, line 33: at Dome C we used daily values or monthly values and obtained a slope 0.49‰∘C. Not sure what you have used here, please explain.*

We also used daily data to compute the correlation between precipitation $\delta^{18}$O and temperature at Dome C. We specified it as follows (p.11, l.33 – p.12 l.2):

"At Dome C, over 3 years, daily precipitation $\delta^{18}$O displays a significant positive correlation with daily AWS near surface air temperature, with a slope of 0.49 ‰ per °C (r=0.79 and p=3.2.E-109). "

*Page 12, line 1: please check the reported statistical values. Also there seem to be a typo error*

We thank Dr Stenni for identifying a mistake in the reported p-value. The new values will be found in p.12, l.1-2:"with a slope of 0.49 ‰ per °C (r=0.79 and p=3.2.E-109)"

*Page 12, line 2-3: at Dome C we considered daily values, so we did not exclude the seasonal cycle. We also obtained a higher slope (1.4) if considering the only 3 annual values, but its significance is low since it is calculated on 3 years only. Here, on the other hand, you are considering inter-annual variability.*

We agree that the $\delta^{18}$O-temperature slope at Dome C is quite variable depending on the considered time scale, and that the inter-annual variability is relevant for the comparison with results from the S1C1 core.

*Page 12, line 14: rather than metamorphism I would say exchanges between surface snow and water vapour.*

We have reformulated the expression (p.12, l.10): "isotopic exchanges between surface snow an water vapour in-between snowfall events".

*Page 12, line 26: the first reported values (r=-0.48 and p=8.0E-3) seem significant. Please, check.*

We have checked the correlation coefficient and the p-value. Indeed, they are significant. However the correlation coefficient is too low to assume a trend.

We have rewritten "Weak but significant decreasing long-term trends appear in both the S1C1 record and the model outputs" (r=-0.38 and p=4.0E-2and r=-0.48 and p=8.4E-3 respectively)." (p.12, l.21-22).

*Page 13, line 21-22: in Antarctica this problem was nicely shown by Frezzotti et al, 2007 JGR. Also for snow accumulation rate values as those found by the authors for this Adelie Land site S1C1 and also considering the wind effects (see comment by E. Isaksson)!*

We have informed the reader of this feature in the introduction. We have first given more information about the winds (cf p. 3 l. 13-16): "These features lead to dominant southerly (160 plus or minus 20°) intense katabatic winds (Wendler et al., 1997). During summer open sea conditions, a reverse pattern of wind direction can occur, associated with sea breezes. At

Dumont d'Urville station (hereafter DDU), Périard and Pettré (1993) reported an average wind speed of 10 m s$^{-1}$, as well as events when wind speeds exceed 20 m s$^{-1}$ during several days (with an observed local maximum wind speed at 90 m s$^{-1}$)."

We have also reported *Frezzoti et al., 2007* in the introduction to report this previous study which show the impact of the high variability in accumulation (due to wind-driven deposition and erosion processes in this region) over ice core records.

5   (p.3, l.17-20): "The resulting high spatial variability in accumulation challenges the climatic interpretation of a single ice core record, especially for accumulation rates lower than 20 cm w.e. y$^{-1}$ (Frezzotti et al., 2007).")

*Page 13, line 27: add "scale" after "horizontal".*
Done.

*Page 14, line 15: the year should be 1986 and not 1985.*
Done

*Page 15, line 7: add a space between "the" and "period".*
15   Done

*Page 15, line 10: what do you mean by "climatic deposition signal" . . . not clear.*
When writing "climatic deposition signal", we meant that the signal represented the synoptic climatic. Changes towards this meaning were made (p.15, l.2): "This finding shows that the nudged ECHAM5-wiso simulation is able to capture almost

20   40% of the inter-annual average accumulation variance, and that there is a clear synoptic climatic deposition (and not only local deposition) in the stake accumulation record".

*Page 16, line 3: why "his"?*
It was a typing error.

*Page 16, line 6-7: comparison with the ECHAM5-wiso output BUT with which chronology?*
It was originally with the first one. But it has changed since we had to correct it. But we should not have referred to this correlation since the climatic signal has been shown thanks to the comparison of the model with the stake data (cf. last correction). Therefore, it was corrected (p.16, l.12-13): "to confirm our hypothesis that there is a significant climatic signal

30   in δ$^{18}$O from this single core, as suggested by the comparison of the average stake area record with the ECHAM5-wiso output."

*Page 16, line 9: contradictory: may you explain better?*
Our initial formulation was confusing and we have removed this sentence.

*Page 16, line 20: Stenni et al., 2016.*
Done

5  *Page 16, line 21: see also the conclusions by Schlosser et al., 2016 (ACP) about the inter-annual difference.*
We have cited this manuscript that is indeed relevant for our work (p. 16, l.28):
"These model results suggest that processes other than local temperature play a key role in local $\delta^{18}O$ seasonal cycle and inter-annual variability, as suggested by Schlosser et al. (2016)."

10  *Page 16, line 28: if I am not wrong, this statement "but a significant and weaker correlation between δ18O from the S1C1 core and simulated by ECHAM5- wiso" depends on the age scale considered, isn't it?*
After correcting the chronology and reprocessing the statistical calculations, we have revised our statements to (p.17, l.4-7) "a high and significant correlation between inter-annual variations in temperature from the READER database and ECHAM5-wiso simulated near surface air temperature (r=0.91 and p=2.2E-12) but no significant correlation between inter-
15  annual variations in $\delta^{18}O$ from the S1C1 core (from the first dating) and simulated by ECHAM5-wiso."

*Page 17, line 10: this adjective "coherent" seems in contradiction with what you have just said few lines before.*
We deleted this adjective.

20  *Page 18, line 4: change sake into stake*
Done

*Page 18, line 7: please consider that the difference in the snow accumulation rate (and wind action) at the different sites and their effects on the diffusion effects and so on the smaller amplitude of the seasonal isotopic cycle. See comment before.*
25  We completed this sentence by the following (p.18, l.17): "But note that the accumulation rate is site-dependent."

*Page 18, line 10-12: the sentence "We stress … relationship." seems not valid if I consider δ core and Techam. May you check?*
The underestimation of the inter-annual variability in $\delta^{18}O$ by ECHAM5-wiso explains its underestimation of the strength of
30  the slope $\delta^{18}O$-temperature. Indeed, in table 4, the slopes for the period 1979-2006 are weaker for $\delta^{18}O$ extracted from the model than estimated using data from S1C1 and READER.  (See table below).

| Resolution | Source of the temperature | Source of the $\delta^{18}O$ | Slope | r | p |
|---|---|---|---|---|---|

| | | | | | |
|---|---|---|---|---|---|
| annual | ECHAM | ECHAM | 0.29 | 0.39 | 3.8E-2 |
| annual | READER database | S1C1,2 | 0.42 | 0.48 | 1.0E-2 |

*Page 18, line 17: I would also add that precipitation sampling at DDU would be desirable.*

We have added that not only vapour isotopic composition measurements at DDU were needed, but also precipitation ones (p.18, l.25-27): "…obtaining continuous vapour and precipitation isotopic composition measurements at DDU will offer the possibility of a direct comparison with atmospheric transport pathways, and with aerosol chemical data."

*Page 18, line 34: the reference Ahmed is not correct! Please change into "PAGES 2k Consortium, 2013". And also Jones et al. is published.*

Done

*Page 19, line 5: please specify what is PSA2.*

We have given its definition (p.19 l.15-16): "PSA2 (associated with the tropical heating anomalies in the Pacific)"

*References: some are not completed (Jones et al, Lemeur submitted, Masson Delmotte et al., 2008 the authors list is not completed) and some must be changed from discussion to final accepted papers (Ritter, Schlosser). Please, make a careful check.*

Done

*Table 1. The precipitation column is not only precipitation (ok for ECHAM) but in the case of the core data is accumulation. What is u? I suppose average. ...*

μ is indeed our notation for the mean value. It has been specified in the legend.

Indeed, ECHAM5-wiso simulates precipitation values, but we have accumulation data for the S1C1 core. It has been clarified in the label and in the legend.

*Table 2: be careful about the year 2007 (the period covered by the core). ... see my comment above (page 2, line 3).*

All tables and figures were corrected.

*Table 3: something is lacking in the first column: You should have 1956-2006 (annual and decadal) and then the same for 1979-2006.*

We readjusted the content of the first column to be placed at the top of each cell.

*Why you did not discuss in the text the relationship between Techam and δ18O S1C11 and S1C12, if I am not wrong? (see my comment above Page 18, line 10-12).*

Relationships using the two chronologies were recalculated. the statistical analysis of linear correlation between READER data and S1C1 d18O leads to no significant result using the first chronology, while it appears significant (but weak) with the revised chronology, and, in this case, with a slope close to the one expected from a Rayleigh distillation (Table 4). So we completed as follows (p. 16 l.22-23):

"We note that over the period 1979-2006, although the $\delta^{18}$O-temperature relationship from the fist chronology is not significant, the relationship from the second one is significant but weak, with a slope close to the Rayleigh distillation."

*Figure 1: add "hatched area" in the caption after "Passive Microwave Data".*

We added "hatched area" after "Passive Microwave Data" as suggested.

*Figure 2: in the caption: you do not have dimethylsulfure data but MSA data!! I would also make the labels larger than they are.*

It was a mistake to write "Dimethyl… (DMS)", and this has been corrected to MSA. Also the figure was re-drawn regarding the corrected alternative chronology.

*Figure 4: the different lines are not distinguishable. Also in this case I would use larger labels. In the caption: resampled data: specify with which step. The annual mean is calculated by the annual layer dating? Specify.*

We have re-drawn this figure, removing the resampled data, which were not distinguishable with the raw data. The annual mean values are based from the resampled data and the accumulation data from the annual layer counting. This information is now provided in the legend.

*Figure 5: Also in this case I would use larger labels. In the caption, some typos. ...*

Larger labels were drawn.

*Figure 6: I had some difficulty in comparing the values referring to the red (cityscape) and the blue lines.*

All data were draw using cityscape lines.

*Moreover, check also here the correct period covered by the core. The first year is 2006, the accumulation record seems OK but for the δ18O why you have a value for 2007? Here the dimensions of the different labels are OK.*

Done

*Figure 7: Also in this case I would use larger labels. Figure 8: Also in this case I would use larger labels. Also here the first δ18O seems to be 2007. . .. To be checked.*

Done

**Author'changes in manuscript**

In this part, we summarize the main corrections implemented in the revised manuscript. After reporting the general corrections which have impacts through all the manuscript, we mention corrections which are specific to each section.

**General corrections:**

- Some sentences were shortened or deleted to make the manuscript shorter.
- Omissions were corrected.
- Some reformulations were made.
- We had made a mistake when copying/pasting the resampled $\delta^{18}O$ versus year and had shifted all dates by one year for both $\delta^{18}O$ and accumulation data. So the chronology was reported to range from 1947 to 2007 instead of 1946-2006. Therefore, we first changed through all the paper the covered period by the S1C1 core. Then, in the following of the study, the selected period for all the climatic records was also the period 1947-2007. We thus have corrected our data (cf. Supplementary Material), and re-processed all the statistical calculations. The main changes were the absence of significant correlation between $\delta^{18}O$ from the S1C1 core (1st dating) and the DDU near surface temperature at the inter-annual scale, and the absence of any significant correlation between $\delta^{18}O$ from the S1C1 core (1st dating) and the SAM or the Niño index. Changes were made in the body of the paper, in the tables (tables 2 and 3) and in the Supplementary Material. Also we have changed the alternative dating (based on the comparison between $\delta^{18}O$ from the S1C1 core and $\delta^{18}O$ outputs simulated by ECHAM5-wiso). Instead of one-year shift, this alternative dating had two-year shift compared to the first one. Details from the choice of the added/deleted identified annual layer is detailed in Section 4.1.

**Title page**: The affiliation associated with LSCE was corrected as it is now part of Université Paris Saclay.

**Abstract:** We changed information linked to the dating which has been corrected:

- The covered period by the S1C1 core: 1946-2006
- The correlation between the $\delta^{18}O$ core from S1C1 and the DDU near surface temperature which is now significant at the annual scale anymore, but only at the decadal scale.
- The alternative dating is now shifted of 2 years.
- We also reformulated some expressions.

1. **Introduction:** The paragraph describing the meteorology of the Adélie Land was completed in order to give useful information, e.g. wind direction and speed wind linked with wind-driven deposition.

2. **Material and method**

**2.1 Field work and ice core sampling**

We reformulated the sentence linked to the comparison of the firn temperature at DDU with summer Dome C and DDU temperatures over the period 1995-2015.

**2.5 Atmospheric simulations**

A number of references had been omitted in the bibliography. The list of references has been updated.

3. **Results**

**3.1.3 Combining layer counting with absolute age horizons**

Details associated with equivocal annual layer identification was deleted and reported in a table (Table 1).

**3.2 Record of accumulation variability and comparison with stake measurements**

Statistical values and spectral analyses of the S1C1 accumulation record were corrected (still because of the dating mistake)

**3.3 Record of $\delta^{18}O$**

**3.3.1 Other sources of information for $\delta^{18}O$ regional variability**

As for accumulation records, statistical values and spectral analyses of $\delta^{18}O$ from the S1C1 core were corrected (still because of the dating mistake)

**3.4 Model-data comparison**

**3.4.1 Comparison between the ECHAM5-wiso and the S1C1 data**

Statistical values between ECHAM5-wiso outputs and S1C1 records were corrected.

4. **Discussion**

**4.1 An alternative age scale can improve the model-data comparison for $\delta^{18}O$**

The identification of summer layers leading to the establishment of the chronology was updated considering a corrected alternative dating, based on 2-year shift compared to the initial chronology.

Correlations using the corrected alternative dating were updated too.

**4.2 Processes causing non-climatic noise**

More information about the representativeness of the spatial accumulation around the S1C1 site was given, based on stake records. We mentioned the high variability of accumulation in the region.

Also, a brief discussion about refreezing, melting and liquid precipitation has been added.

**4.4 $\delta^{18}O$-temperature relationship**

All correlations using the S1C1 records (from both dating) were corrected.

**5. Conclusions and perspectives**

Relationships between SAM and el Nino index with $\delta^{18}O$ from the S1C1 core were recalculated and found to be insignificant.

[revised manuscript text omitted]

---

## Author Response (AR2)

Dear editor,

We thank you for the acceptation of our paper for the publication in the Cryosphere. We also wish you a happy new year 2017. You will find below answers to your comments and suggestions, and a marked-up version of the manuscript.

**Reply to the editor's comments**

*Dear dr. Goursaud,*

10   *thank you for submitting your revised manuscript.*

*Pending some technical, linguistic corrections listed below, I am happy to report that your paper is now accepted for publication in The Cryosphere.*

15   *All the best for 2017,*

*Michiel van den Broeke*

20   *Textual corrections*

*p. 2, l. 10: Remove 'data'.*
Done.

25   *p. 3, l. 3: please replace 'Consortium' by "PAGES 2k Consortium'; here and in reference list.*
Done.

*p. 3, l. 5: remain -> remains*
Done.

*p. 3, l. 11: a large -> large*
Done.

*p. 3, l. 31: thanks to a -> thanks to*

Done.

*p. 4, l. 1: Stenni et al, 2016). And -> Stenni et al, 2016), and*

Done.

*p. 4, l. 5: due to its -> for its*

Done.

*p. 5, l. 5: remove "punctual"*

Done.

*p. 8, l. 23: were -> was*

Done.

*p. 8, l. 24: Antarctic -> Antarctica*

Done.

*p. 9, l. 32: None of these -> These*

Done.

*p. 11, l. 29: observe -> we observe*

Done.

*p. 12, l, 22: type "outputsin"*

Done.

*p. 12, l. 23 and throughout the MS: p=2.2E-2 -> p = 0.022*

Done.

*p. 16, l. 33: O.63 -> 0.63*

Done.

*p. 17, l. 7: twice smaller than -> two times smaller than OR twice as small as*

Done.

5  *Together, Sections 4 and 5 contain too much overlap, are too long and therefore cumbersome to read. Please go through Section 5 carefully and eliminate all text that is not strictly necessary. Avoid repetition.*

In order to shorten the section and be more concise, here are the changes we made:

1) p.18 l.12: we removed the periodicities of the records from the S1C1 core.
2) p.18 l.16: we do not precise anymore that wind scouring is linked to katabatic winds.
10  3) P.18 l.24: we do not stress the implication of the underestimation of the inter-annual variability in ECHAM5-wiso $\delta^{18}O$ outputs over the simulated $\delta^{18}O$-temperature slope anymore.
4) p.18 l.26: we do not precise the type of data to compare with the model outputs.
5) p19 l.6: here, we do not summarize the use of the ECHAM5-wiso model anymore. We just add what were the other use of the model than detecting the seasonal cycle in precipitation $\delta^{18}O$ (what was written before in the section).

**Author's changes in the manuscript**

To summary, linguistic mistakes were corrected following the list you made. Also, p-values written in its scientific notation, with ten to the power of minus 2, were written in their decimal notation. Finally, we shortened the fifth section, removing what we considered as not necessary.

[revised manuscript text omitted]